# Selective rab11 transport and the intrinsic regenerative ability of CNS axons

Hiroaki Koseki[1,2], Matteo Donegá[2], Brian YH Lam[3], Veselina Petrova[1,2], Susan van Erp[4], Giles SH Yeo[3], Jessica CF Kwok[1,2,5,6], Charles ffrench-Constant[4], Richard Eva[1,2]*, James W Fawcett[1,2,6]*

[1]John van Geest Centre for Brain Repair, University of Cambridge, Cambridge, United Kingdom; [2]Department of Clinical Neurosciences, University of Cambridge, Cambridge, United Kingdom; [3]MRC Metabolic Diseases Unit, Metabolic Research Laboratories, University of Cambridge, Cambridge, United Kingdom; [4]MRC Centre of Regenerative Medicine, University of Edinburgh, Edinburgh, United Kingdom; [5]School of Biomedical Sciences, Faculty of Biological Sciences, University of Leeds, Leeds, United Kingdom; [6]Centre of Reconstructive Neuroscience, Institute of Experimental Medicine, Czech Academy of Sciences, Prague, Czech Republic

**Abstract** Neurons lose intrinsic axon regenerative ability with maturation, but the mechanism remains unclear. Using an in-vitro laser axotomy model, we show a progressive decline in the ability of cut CNS axons to form a new growth cone and then elongate. Failure of regeneration was associated with increased retraction after axotomy. Transportation into axons becomes selective with maturation; we hypothesized that selective exclusion of molecules needed for growth may contribute to regeneration decline. With neuronal maturity rab11 vesicles (which carry many molecules involved in axon growth) became selectively targeted to the somatodendritic compartment and excluded from axons by predominant retrograde transport However, on overexpression rab11 was mistrafficked into proximal axons, and these axons showed less retraction and enhanced regeneration after axotomy. These results suggest that the decline of intrinsic axon regenerative ability is associated with selective exclusion of key molecules, and that manipulation of transport can enhance regeneration.

DOI: https://doi.org/10.7554/eLife.26956.001

*For correspondence:
re263@cam.ac.uk (RE);
jf108@cam.ac.uk (JWF)

**Competing interests:** The authors declare that no competing interests exist.

## Introduction

Axon regeneration fails in the adult mammalian CNS due to a combination of extrinsic inhibitory cues and an inadequate intrinsic regenerative response (*Fawcett et al., 2012*; *Liu et al., 2011*). Long-distance regeneration can only be achieved if axons have a high intrinsic growth ability (*Liu et al., 2010*; *Cheah et al., 2016*). Embryonic axons have this ability, and can elongate for long distances if immature neurons are transplanted into the adult CNS environment (*Lu et al., 2012*; *Reier et al., 1986*). However with maturation, adult CNS neurons lose much of this intrinsic regeneration ability, and axons such as those of the corticospinal tract show limited growth even in a permissive environment such as a peripheral nerve graft (*Bradke et al., 2012*; *Geoffroy et al., 2016*) (*Richardson et al., 1984*). Several changes occur during neuronal differentiation that might explain this maturational reduction in growth ability. Amongst these a key factor that changes radically with maturity is the establishment of selective transport to axons and dendrites (*Bentley and Banker, 2016*; *Britt et al., 2016*; *Franssen et al., 2015*; *Maeder et al., 2014*; *Petersen et al., 2014*). Polarised transport of proteins is required for the correct molecules to travel to pre-and postsynaptic sites and in order for axons and dendrites to possess different structures and functions (*Britt et al., 2016*;

**eLife digest** The nerves in the brain and spinal cord can be damaged by trauma, stroke and other conditions. Damage to these nerve fibres can destroy the connections they form with each other, which may lead to paralysis, loss of sensation and loss of body control. If we could stimulate the regeneration and reconnection of the damaged nerve fibres then neurological function could be restored. However, although embryonic nerve fibres can regenerate when they are transplanted into the adult central nervous system, this regenerative ability appears to be lost as the nerve fibres mature.

To investigate when and why nerve fibres lose the ability to regenerate, Koseki et al. first developed a tissue culture assay in which individual nerve fibres were cut with a laser and imaged for several hours to track their regeneration (or failure to regenerate). The results demonstrate that nerve fibres from the central nervous system progressively lose the ability to grow and regenerate as they mature.

To investigate why mature nerve fibres cannot regenerate, Koseki et al. measured whether nerve fibres can transport some of the molecules needed for growth and regeneration to sites of damage. This showed that the compartments in which some key growth molecules are transported become excluded from mature nerve fibres. These compartments are marked by a protein called rab11, and Koseki et al. found that forcing rab11 back into mature nerve fibres restored their ability to regenerate.

There is still a lot of work needed before these findings can lead to a new regeneration treatment for patients, but it is a crucial step forwards. Furthermore, the assay developed by Koseki et al. could be used to develop and test such treatments.

DOI: https://doi.org/10.7554/eLife.26956.002

*Maeder et al., 2014*). A consequence of polarised transport is that several key growth-related molecules including integrins, trkB and IGF receptor become excluded from cortical axons as they mature (*Andrews et al., 2016*; *Franssen et al., 2015*; *Hollis et al., 2009a*; *Hollis et al., 2009b*). This development of polarised transport therefore provides a possible mechanism for the maturational loss of intrinsic regeneration ability. Our previous work has focused on integrins as an example of a molecule necessary for efficient regeneration and growth (*Andrews et al., 2009*; *Tan et al., 2011*; *Cheah et al., 2016*), and has shown that these receptors are progressively excluded from cortical axons to become exclusively somatodendritic both in vitro and in vivo (*Franssen et al., 2015*; *Andrews et al., 2016*). Conversely, sensory and retinal ganglion cell neurons, which can be more easily manipulated to make them regenerate, continue to transport integrins into their axons into adulthood (*Andrews et al., 2016*).

Integrins are transported into axons in recycling endosomes marked by the small GTPases rab11 and ARF6 (*Eva et al., 2010*; *Eva et al., 2012*; *Franssen et al., 2015*). Rab11 is implicated in the trafficking of a number of growth associated molecules, including Trks A and B (*Ascaño et al., 2009*). Integrins, Trks and rab11 are all excluded from cortical CNS axons in-vivo (*Andrews et al., 2016*; *Hollis et al., 2009a*; *Sheehan et al., 1996*). Rab11 is involved in axon elongation in young CNS and adult PNS neurons regenerating in vitro, and is required for correct growth cone function as its targeted removal leads to growth cone collapse (*van Bergeijk et al., 2015*; *Eva et al., 2010*). These findings led us to ask whether selective exclusion from axons of the rab11 vesicles that transport these growth-related molecules contributes to their maturational loss of intrinsic regenerative ability, and whether replenishment and/or modulation of activation of rab11 can enable regeneration.

Previous studies of selective transport have used an in vitro model in which cortical neurons mature and exclude integrins from their axons (*Franssen et al., 2015*; *Petersen et al., 2014*). We further developed this culture model for regeneration studies, and demonstrated that axons in these cultures lose the ability to regenerate with maturity, and that this loss is intrinsic to the axons rather than due to environment. We found that the characteristics of retraction bulb formation define the probability of regenerating a growth cone, but that subsequent axon elongation is controlled differently. The behaviour of rab11 was then investigated: we find that it becomes excluded from axons

as neurons mature in-vitro, but that restoring the presence of rab11 in axons modifies retraction and enhances regeneration of mature axons.

## Results

### Cortical neurons mature in the in-vitro culture model

Embryonic CNS neurons can differentiate in culture and provide a model for maturation-related changes (*Barbati et al., 2013*). We asked whether this type of culture could also be used to model the developmental loss of intrinsic regenerative capacity. Dissociated embryonic day 18 (E18) rat cortical neurons were grown in the presence of astrocyte feeder cultures (*Kaech and Banker, 2006*). Neuronal maturity was tracked by examining the electrical properties of neurons, their pattern of gene expression through mRNA sequencing, and immunolabelling of cytoskeletal maturity markers. Electrophysiological maturation was examined by patch clamping neurons at 4, 8, 16, and 24 days in vitro (DIV) (*Figure 1*, *Figure 1—figure supplement 1*). During time in culture, resting membrane potential lowered to −55 mV (*Figure 1A*) and membrane capacitance increased (*Figure 1B*). Only 29.4 ± 10.6% of neurons were able to fire action potentials in response to steps of depolarizing current at 4 DIV, but the percentage increased to 100% by 24 DIV (*Figure 1C,E*). Concurrently the action potential spikes turned from single to multiple (*Figure 1E*), increased in amplitude (*Figure 1— figure supplement 1A*), decreased in duration (*Figure 1—figure supplement 1B*) and spike

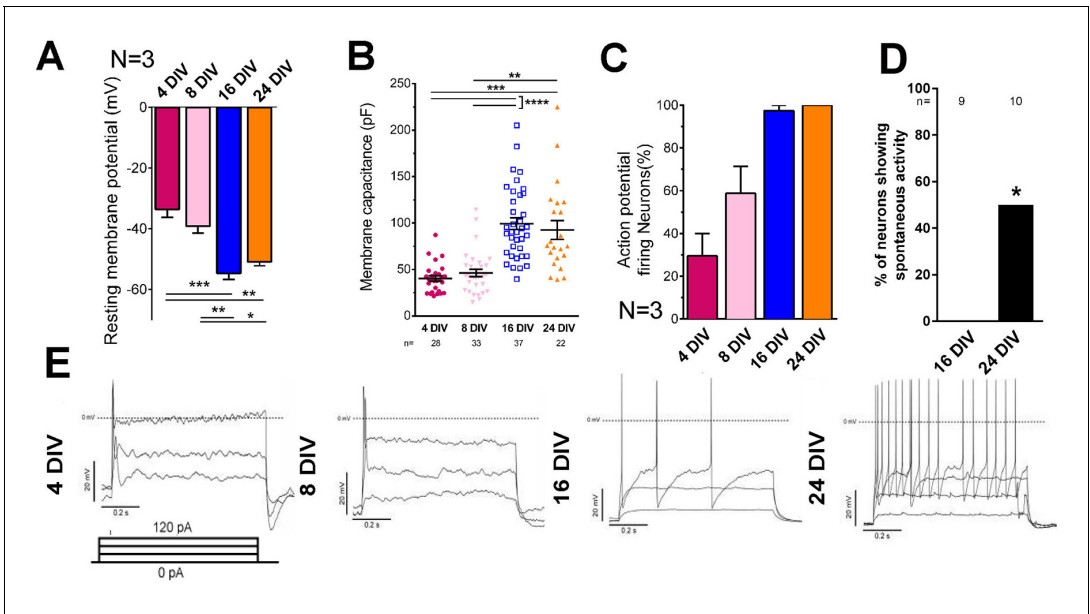

**Figure 1.** Electrical maturation of neurons. (**A**) Resting membrane potential of maturing neurons. The average membrane potential from at least three patching sessions is shown. At least 10 neurons per session. One-way ANOVA followed by Bonferroni's post hoc test. (**B**) Membrane capacitance of maturing neurons. One-way ANOVA followed by Games-Howell post hoc test. (**C**) Percentage of patched neurons capable of firing action potentials when depolarizing currents were applied. The average percentage from at least three patching sessions is shown. At least 10 neurons/session. At 24 DIV, all neurons fired action potentials. Representative responses are shown in (**E**). (**D**) Percentage of neurons showing spontaneous activity. Fisher's exact test. Error bars represent s.e.m. Patch clamp results from at least three independent sessions were accumulated. Sample numbers are shown in the figure. *p<0.05, **p<0.01, ***p<0.001 and ****p<0.0001.

DOI: https://doi.org/10.7554/eLife.26956.003

The following figure supplements are available for figure 1:

**Figure supplement 1.** Electrical and anatomical characterisation of maturing neurons.
DOI: https://doi.org/10.7554/eLife.26956.004

**Figure supplement 2.** Characterisation of mRNA and protein expression changes with maturity.
DOI: https://doi.org/10.7554/eLife.26956.005

**Figure supplement 3.** Confirmation of axon initial segment development and neuronal viability.
DOI: https://doi.org/10.7554/eLife.26956.006

threshold increased (*Figure 1—figure supplement 1D*) while input resistance decreased (*Figure 1—figure supplement 1E*). All of these are changes associated with the maturation of neurons in vivo. Interestingly, the main statistical difference for these parameters was observed between 8 DIV and 16 DIV. As for network development, compatible with a previous study (*Kay et al., 2011*), co-localization of both excitatory and inhibitory synaptic markers was observable by 10DIV (*Figure 1—figure supplement 1F*). However, spontaneous activity increased between 16 DIV and 24 DIV (*Figure 1D*).

RNA sequencing showed progressive changes in many genes towards expression patterns typical of mature neurons. From the mRNA expression data we picked two useful maturity markers which are cytoskeletal-related molecules and show robust immunostaining: doublecortin and low molecular weight neurofilament. Immunostaining showed a decrease in doublecortin and an increase in low molecular weight neurofilament with neuronal maturation similar to the changes in their mRNA levels (*Figure 1—figure supplement 2A,B,C*). Ingenuity pathway analysis of genes with increasing expression showed many changes in molecules involved in synapse formation (*Figure 1—figure supplement 2D*), whereas many of the genes with decreasing expression were involved in neuronal development (*Figure 1—figure supplement 2E*). Full results are available on NCBI GEO DataSets (RRID:SCR_005012, accession no: GSE92856). Neurons started to develop axon initial segments visualized by immunostaining for neurofascin and Ankyrin-G as early as 3 DIV, and the structure consolidated with increasing DIV (*Figure 1—figure supplement 3A*). There was little neuronal loss in the cultures up to DIV24, and neurons showed no cleaved caspase 3 expression after 3 weeks in culture (*Figure 1—figure supplement 3B,C*). These results demonstrate that neurons in this culture model show maturational changes similar to those of neurons in vivo and remain viable.

## Characterization of the axotomy response of rat cortical neurons

In vitro laser axotomy was performed at 4, 16 and 24 DIV and the events that follow axotomy were recorded. To distinguish individual axons from the large number of surrounding processes, small numbers of neurons were transfected with GFP and before axotomy fluorescent live neurofascin staining of the axon initial segment (AIS) was used in all cases to identify axons from dendrites (*Figure 1—figure supplement 3A*). Events after axotomy unfold over several hours for CNS axons (compared with sensory axons which usually regenerate within one hour). Cultures were therefore imaged using time-lapse microscopy for 10 ~ 20 hr. Axotomy was followed by either neuronal death, branch loss or retraction bulb formation (*Figure 2A,B*). Neurons were considered dead if axon disintegration or cell disruption was seen within 10 hr of axotomy, and were excluded from further analysis: this occurred in 15 ~ 25% of axotomies with no variation with DIV (*Figure 2B,C*). Branch loss was the retraction of the cut axon branch to the nearest branch point with no formation of a retraction bulb: this occurred in 10 ~ 15% of axotomies (*Figure 2B,C*) which were also excluded from further analysis. Retraction bulb formation was the commonest result of axotomy. Starting immediately after axotomy, the GFP signal within the axon disappeared proximal to the cut for up to 1000 μm. Following this, the cytosol would reflux up to the newly sealed axon tip and start accumulating to form a retraction bulb which was usually motile; this usually took from 1 to 3 hr (see below) (*Figure 2A,C*). The appearance of the retraction bulb under fluorescence imaging and phase contrast were identical.

After successful formation of a retraction bulb three outcomes were seen (*Figure 3*); regeneration of a new axon from the bulb (usually following a different track) (*Figure 3A,B*), failure of regeneration (although the retraction bulbs were usually motile), or ectopic growth (formation of a new axon branch from the side of the cut axon, usually within 100 μm of the bulb) (*Figure 3B*). The probability of these events changed radically with neuronal maturity; in 3–5 day old neurons over 70% showed axon regeneration, but by 23–30 days less than 10% regenerated (*Figure 3C*). Typical videos of successful regeneration in a DIV4 neuron and failure of regeneration in a DIV30 neuron are shown in (*Videos 1* and *2*). In the further analysis we have combined the results of regeneration and ectopic growth into an overall regeneration score. To describe regeneration more precisely, we also measured six regenerative factors; retraction distance, retraction bulb formation time, regeneration ratio, regeneration initiation time, regeneration length, and growth cone area and these were measured for proximal and distal axotomies. (measures defined in *Table 1*).

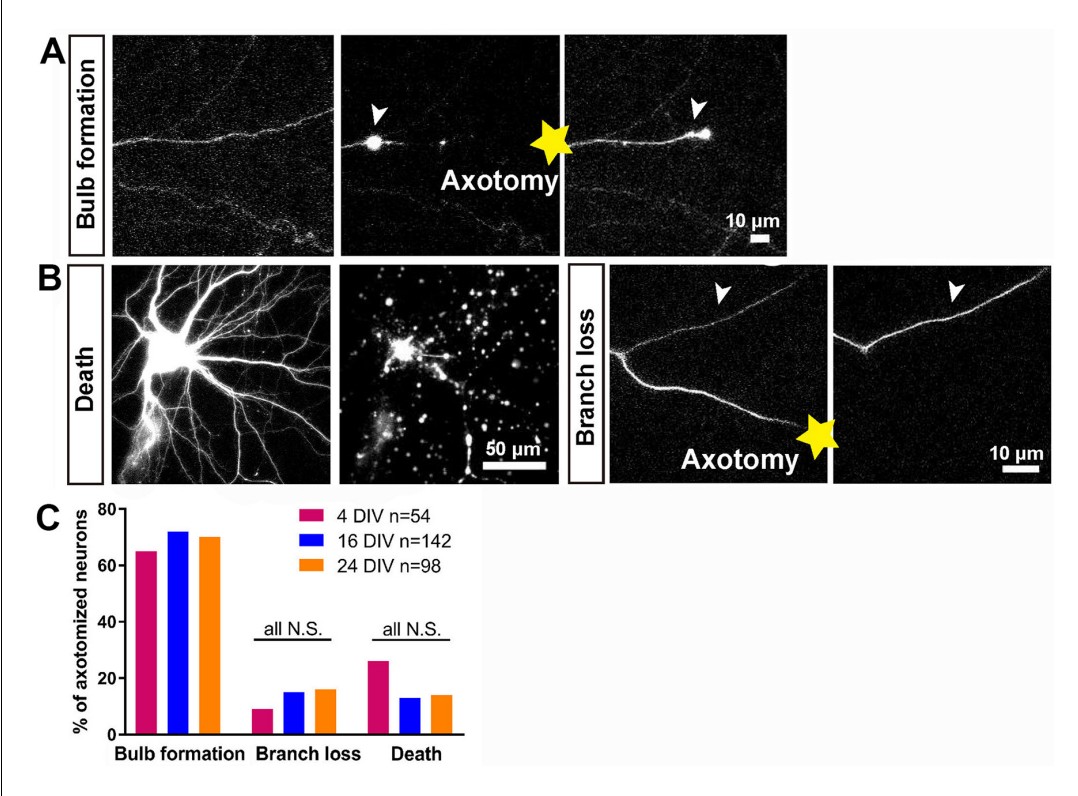

**Figure 2.** Characterisation of the initial response following axotomy. (**A,B**) Representative images of death, branch loss, bulb formation. Axotomy location is indicated by the yellow star. (**C**) Ratio of bulb formation, branch loss and cell death categorized by DIV. Fisher's exact test with Bonferroni correction.

DOI: https://doi.org/10.7554/eLife.26956.007

## Maturation-related changes in retraction and regeneration

The dynamics of retraction bulb formation varied considerably with maturity. The retraction distance increased with DIV (*Figure 4*, *Figure 4—figure supplement 1A*) and the time taken to form a retraction bulb increased from 1.39 ± 0.21 hr in 4 DIV neurons to 2.68 ± 0.20 hr in 24 DIV neurons (p<0.0001, *Figure 4—figure supplement 1B*), correlating with retraction distance (*Figure 4—figure supplement 1C*). The retraction distance had a log Gaussian distribution and we used a log10 conversion for statistical analysis, finding an increase and change in distribution between 4 DIV and 24 DIV (log10 4 DIV: 1.65 ± 0.06, 24 DIV: 2.16 ± 0.03, p<0.0001, *Figure 4A*. The same data are plotted on a non-log scale in *Figure 4—figure supplement 1A–E*). Interestingly a biphasic distribution with long and short retractors could be seen at 16 DIV, suggesting a transition period, and by 24DIV all axons were long retractors. We analysed these long (≧ 70 μm) and short (<70 μm) retracting neurons separately in our further analysis of retraction distance and regeneration. For retraction distances, in 4 DIV neurons and short retracting 16 DIV neurons the position of axotomy was not a factor; short retracting axons were seen after both proximal and distal axotomy (*Figure 4B*). However the in long retracting 16 DIV neurons and all 24 DIV neurons, distal axotomy led to a greater retraction distance on average than proximal axotomy (>600 μm for 16 DIV and >400 μm for 24 DIV) (*Figure 4B Figure 4—figure supplement 1E*). We next focused on regeneration. The DIV of neurons had a powerful negative impact on regeneration, with only 8% of 24 DIV neurons that formed a retraction bulb regenerating compared to 63% in the 4 DIV group (p<0.0001, *Figures 3C* and *4C*). Longer retracting neurons showed a general tendency for less regeneration (*Figure 4G*), consistent with a previous in vivo result (*Canty et al., 2013*). Therefore, we categorized axons according to short or long retraction, and by proximal or distal axotomy (*Figure 4C*). Distal axotomy led to less regeneration in the long retraction group of 16 DIV neurons (proximal 64% vs distal 11%, p=0.0002) and in 24 DIV neurons which all showed long retraction (proximal 24% vs distal 2%, p=0.0116), but

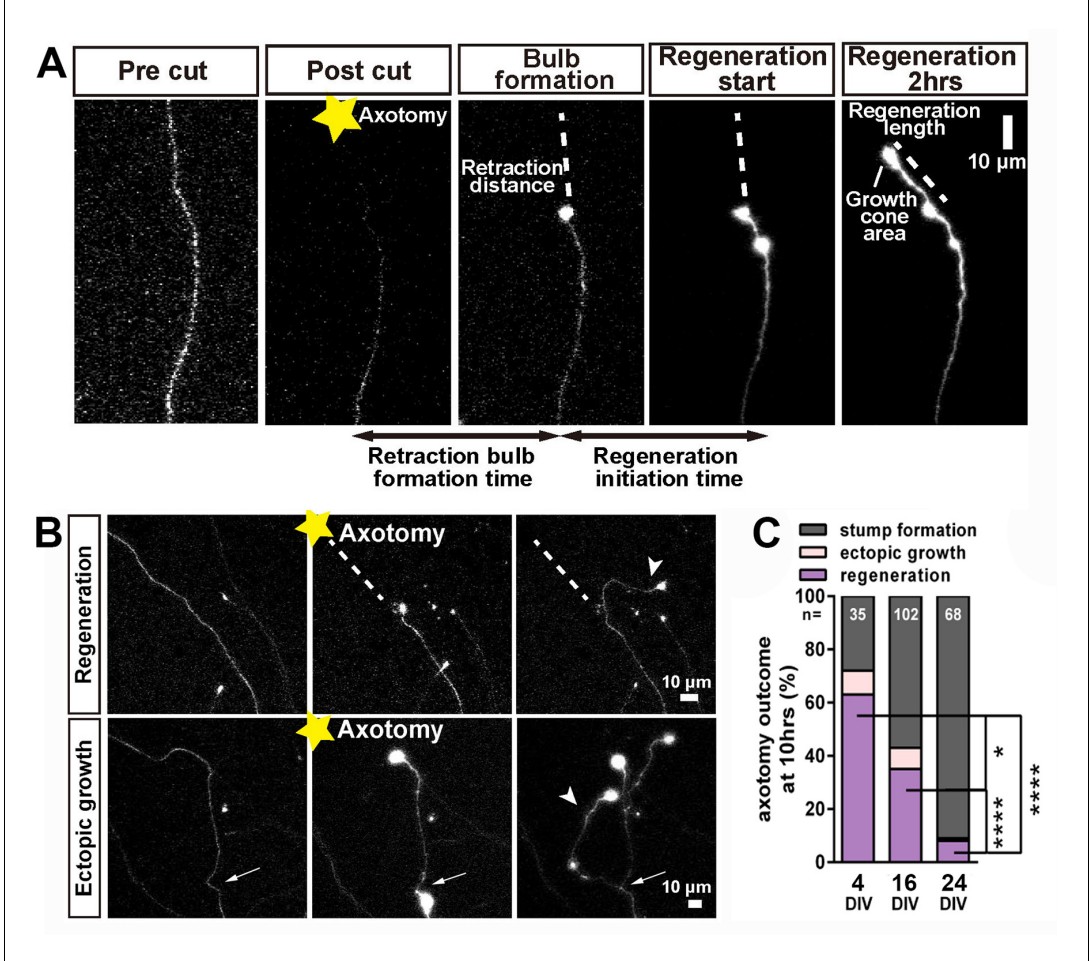

**Figure 3.** Regeneration after axotomy. (A) An axon before cutting, followed by retraction bulb formation and regeneration. (B) A further example of regeneration after axotomy and an example of ectopic growth. (C) The overall proportion of axons showing the different forms of behaviour after axotomy at different stages of maturity. Fisher's exact test with Bonferroni correction. Error bars represent S.E.M.. *p<0.05, ****p<0.0001.
DOI: https://doi.org/10.7554/eLife.26956.008

distal axotomy did not affect regeneration success in the short retraction groups (p=0.0325) (*Figure 4C*). In summary, with increasing DIV retraction distance increased and so did regeneration failure, particularly in axons that showed long retraction and after distal axotomy.

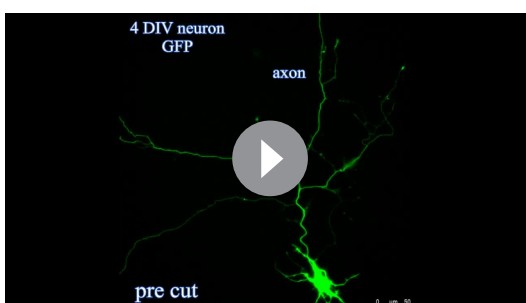

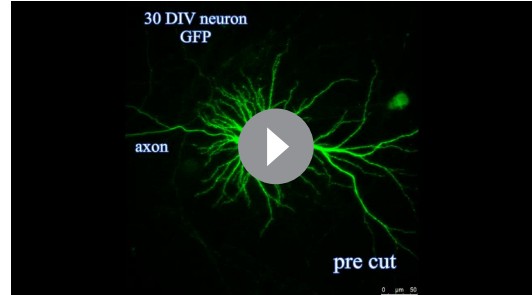

**Video 1.** shows successful regeneration of an axon from an immature neuron, cut at 4 DIV.
DOI: https://doi.org/10.7554/eLife.26956.010

**Video 2.** shows failure of regeneration of an axon from a mature neuron, cut at 30 DIV.
DOI: https://doi.org/10.7554/eLife.26956.011

We examined other growth cone behaviour measurements related to regeneration, including regeneration initiation time which increased with DIV and growth cone area which declined (*Figure 4D,F*). The length of the regenerated axon 2 hr after regeneration initiation also declined with DIV (*Figure 4E*). These factors mainly changed between 4 and 16 DIV, and within the 16 DIV axons, retraction and axotomy distance did not influence these measures (*Figure 4—figure supplement 1F,G,H*). Taken together, it can be concluded that (1) long retraction is prognostic for poorer regeneration initiation, and (2) the neuron's ability to successfully initiate regeneration does not dictate how rapidly the axon will then elongate.

## The maturing environment of the culture is not inhibitory to regeneration

Having established a model which shows regenerative decline with DIV, we asked whether this was due to a build-up of inhibition in the culture environment or to an intrinsic loss of regeneration ability in the axons. To address this, we plated newly harvested immature neurons transfected with GFP onto mature 21 DIV cultures without changing the medium, and allowed them to grow for 4 days, resulting in a 4 DIV neuron surrounded by a 25 DIV environment (*Figure 5*). Interestingly, 4 DIV neurons on 25 DIV cultures had longer and more complex axons than 4 DIV neurons on PDL (*Figure 5A–C*). When axotomized, 4 DIV neurons on 25 DIV cultures behaved very similarly to 4 DIV neurons on PDL, with short retraction (*Figure 5D*), and high regeneration success (54% for 4 DIV neurons on PDL, 46% for 4 DIV neurons on 25 DIV cultures, and 2% for 24 DIV neurons, p<0.0001, *Figure 5E*). These results demonstrate that maturation of the culture environment does not explain the observed regenerative decline with DIV.

## Rab11 positive vesicles are excluded from axons in mature neurons

Having achieved an in vitro model in which to study the loss of intrinsic axon regeneration with maturation, we explored potential mechanisms. Based on our previous work, our hypothesis was based on the selective axon transportation that develops with maturity, directing some molecules to dendrites others to axons. We asked whether this process leads to exclusion of molecules from the axon that are necessary for regeneration. We focused on rab11 because it is associated with the recycling endosomes responsible for bringing growth receptors and integrins into axons, and because these endosomes are reported to be restricted to a somatodendritic distribution in-vivo (*Sheehan et al., 1996*; *Eva et al., 2010*; *2012*). The distribution of endogenous rab11 was studied by immunostaining, which revealed that at 4 DIV rab11 is present equally in both axons and dendrites, but becomes exclusively somatodendritic by 16 DIV (axon stem/dendrite ratio: 4 DIV 2.05 ± 0.21, 16 DIV 0.23 ± 0.03, p=0.0001, *Figure 6A,B*). The immunofluorescence intensity of rab11 in the cell bodies did not change between DIVs (*Figure 6C*), and neither did the mRNA level (*Figure 1—figure supplement 2F*), whereas the axon stem/cell body ratio decreased and dendrite/cell body ratio increased between 4 DIV and 16 DIV (*Figure 4B,D*). Collectively these results show that rab11 becomes progressively excluded from axons but not dendrites as neurons mature.

In order to understand the transport dynamics that lead to the selective distribution of rab11, we examined rab11 trafficking in axons and dendrites at 16 DIV using transfection of GFP-tagged rab11a. Previous studies have demonstrated different trafficking mechanisms for the active and inactive forms of rab11 (*Welz et al., 2014*), so tagged wild type (WT), dominant negative (DN), and constitutively active (CA) forms of rab11a were transfected. Overexpression of all forms caused a degree of mis-trafficking in neurons, leading to some rab11 being found in proximal axons and an increase in the proximal axon/dendrite ratio for total rab11 (GFP 0.62 ± 0.05, WT 0.63 ± 0.04, DN 0.61 ± 0.04, CA 0.51 ± 0.05, *Figure 7A,B*). However, the quantity declined rapidly with distance from the cell body and the transfected rab11 failed to reach distal axons compared to GFP, confirming an active but slightly leaky exclusion mechanism for rab11 vesicles in axons (*Figure 7C,D*). Previously we have shown that integrin-containing vesicles move mostly retrogradely in mature axons (*Franssen et al., 2015*), so we examined rab11 vesicle movements using time lapse imaging, illustrated in kymographs. Similar to integrins, there was a predominance of retrograde movement in 16 DIV axons, and little anterograde vesicle movement. In dendrites, however movement was equal in both directions (*Figure 7E,F*). Predominantly retrograde transport in axons was seen for all rab11 forms, most prominently with the DN (*Figure 7F*). The average vesicle velocity for both anterograde

**Table 1.** Definition of the measures of regeneration

| Regeneration factor | Regeneration factors were measured as below when a neuron was categorized as regeneration |
| --- | --- |
| Retraction distance | The length of axon that was lost between the location of axotomy and the initial retraction bulb |
| Retraction bulb formation time | The time it took to form the initial retraction bulb, measured from the time point of axotomy |
| Regeneration ratio | The number of neurons that regenerated, over the total neurons which formed a retraction bulb |
| Regeneration initiation time | The time between the retraction bulb formation and the start of a steady extension lasting more than 1 hr and leading to regeneration |
| Regeneration length | The length of axon that extended within 2 hr after regeneration initiation time |
| Growth cone area | The average of the extending tip area measured at 20, 40, and 60 min after regeneration initiation |

DOI: https://doi.org/10.7554/eLife.26956.009

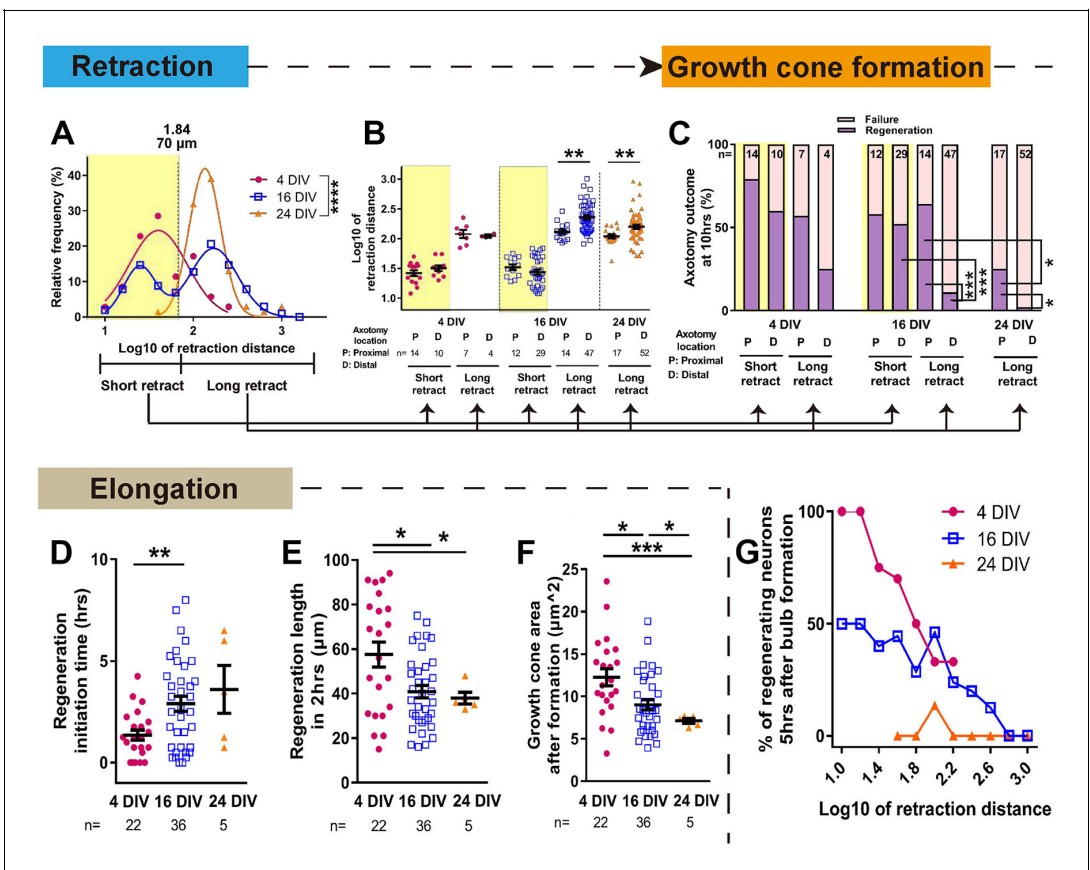

**Figure 4.** Maturation related changes in retraction and regeneration. (A) Relative frequency plot of retraction distance categorized by DIV. Average distance comparison between 4 DIV and 24 DIV: Welch's t-test. (B) Retraction distance categorized separately by DIV, retraction, and axotomy distance. 16 DIV: One-way ANOVA followed by Bonferroni's post hoc test, 24 DIV: Welch's t-test. (C) Regeneration ratio categorized by DIV, retraction, and axotomy distance. 16 DIV: Fisher's exact test with Bonferroni correction, 16 DIV proximal axotomy vs 24 DIV proximal axotomy: Fisher's exact test, 24 DIV: Fisher's exact test. (D,E,F) Regeneration initiation time, regeneration length and growth cone area categorized by DIV. One-way ANOVA followed by Games-Howell post hoc test. (G) Plot of regeneration success against log retraction distance. Error bars represent s.e.m. Axotomy results were accumulated from at least three independent axotomy sessions. Retraction distance was converted by log10. *$p < 0.05$, **$p < 0.01$, ***$p < 0.001$ and ****$p < 0.0001$.

DOI: https://doi.org/10.7554/eLife.26956.012

The following figure supplement is available for figure 4:

**Figure supplement 1.** Further analysis of changes in retraction and regeneration with maturity.

DOI: https://doi.org/10.7554/eLife.26956.013

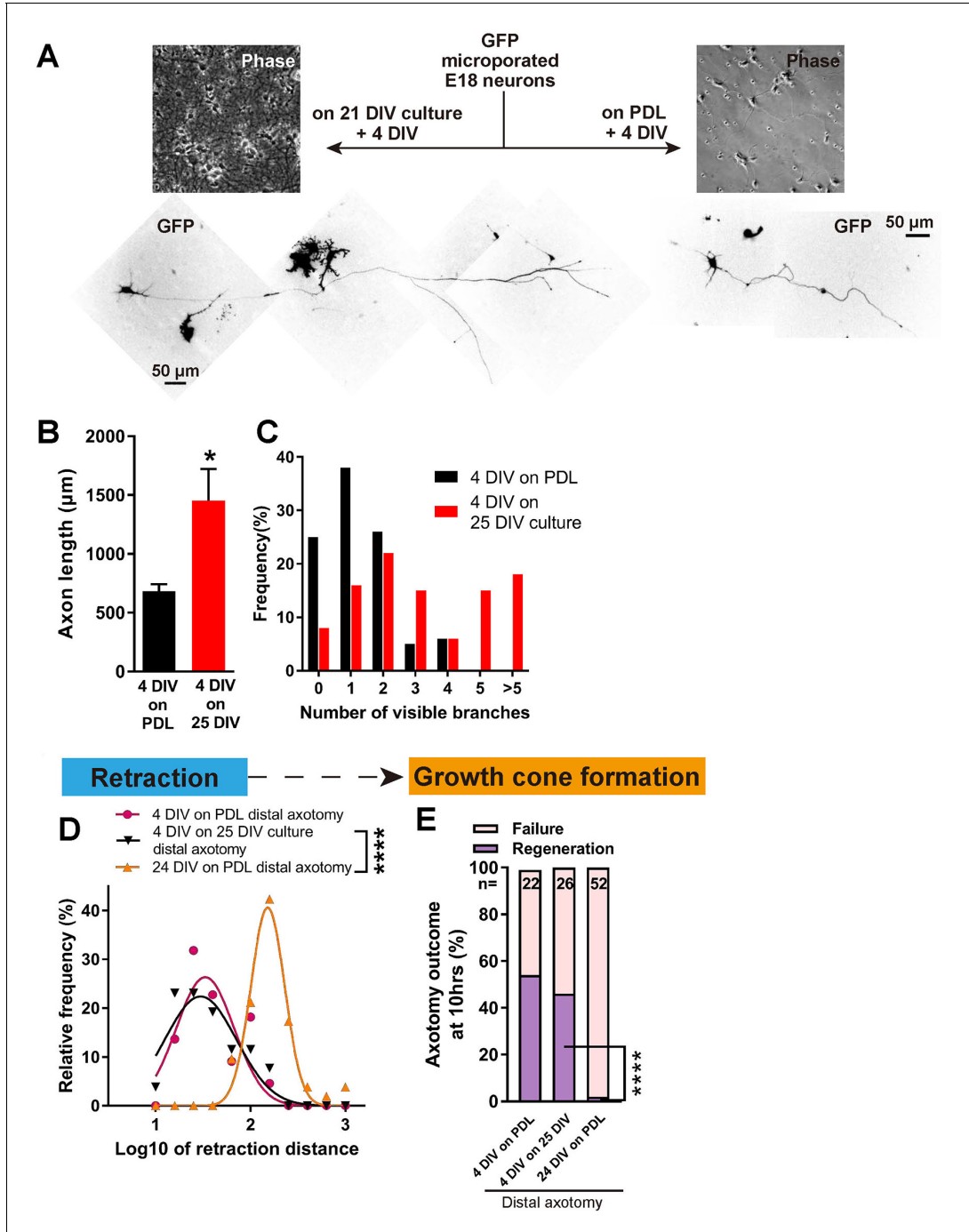

**Figure 5.** Decline in regeneration is not due to the mature environment. (**A**) GFP transfected E18 cortical neurons were plated on either PDL glass or 21 DIV cultures, and cultured for 4 days. (**B,C**) Quantification of axon length (**B**) and branch numbers (**C**) 4 days after plating on PDL or 21 DIV cultures. n = 3 independent cultures, at least 20 neurons/culture. Student's t-test. (**D**) Relative frequency plot of retraction distance categorized by DIV and plating surface. Distal axotomy results were compared. 4 DIV on 25 DIV vs 24 DIV on PDL: Student's t-test. (**E**) Regeneration ratio categorized by DIV and plating surface. Distal axotomy results were compared. Fisher's exact test. One-way ANOVA followed by Games-Howell post hoc test. Error bars represent s.e.m. Retraction distance was converted by log10. *p<0.05, **p<0.01 and ****p<0.0001.

DOI: https://doi.org/10.7554/eLife.26956.014

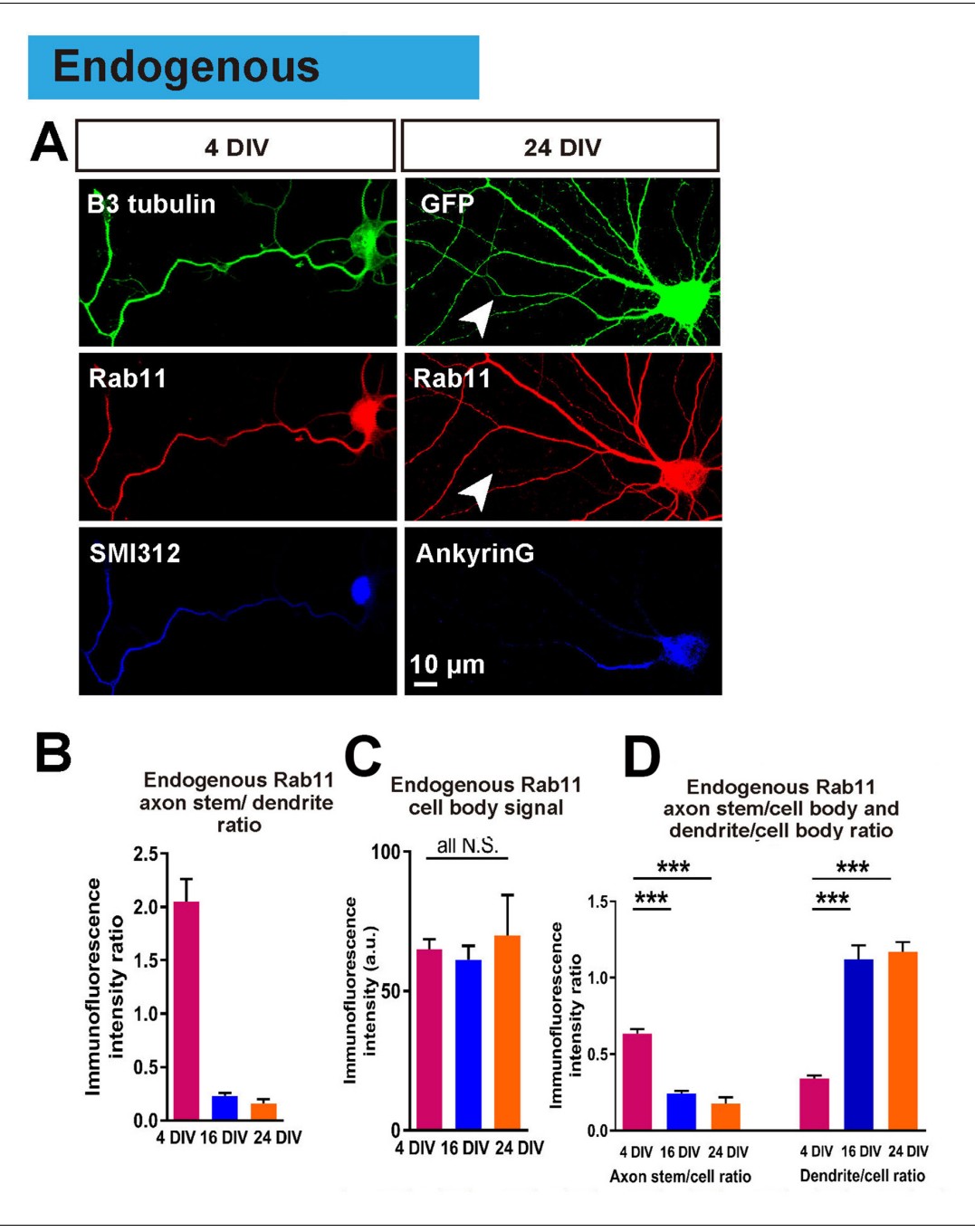

**Figure 6.** Endogenous rab11 becomes restricted to the somatodenritic domain with maturation. (**A**) Immunofluorescence staining of rab11 in 4 and 24 DIV neurons. Neurons were co-stained with beta-3 tubulin or GFP for neuronal outline, and SMI312 or Ankyrin-G for axon marking. (**B,C,D**) Quantification of rab11 immunofluorescence intensity. Immunofluorescence intensities of the cell body, axon stem and proximal dendrite were measured, and ratios were calculated. Staining conditions were kept consistent and images were acquired under the same exposure. n = 3 independent cultures. ANOVA followed by Bonferroni's post hoc test. Error bars represent S.E.M. ***p<0.001.
DOI: https://doi.org/10.7554/eLife.26956.015

and retrograde movement was also slower in axons for WT and CA compared to dendrites (*Figure 7—figure supplement 1A,B*). Taken together, these results show that rab11 becomes selectively excluded from axons by 16 DIV, and that the overall direction of rab11a transport prioritizes exclusion from axons and transport into dendrites by 16 DIV.

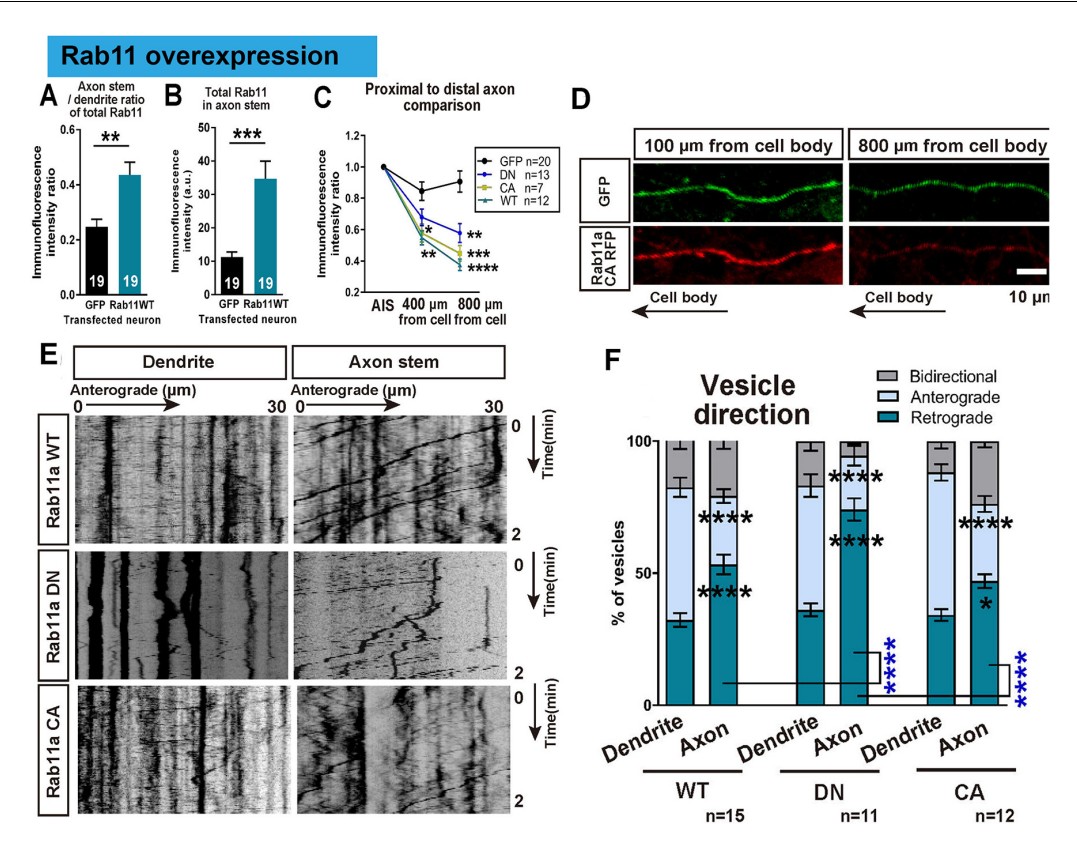

**Figure 7.** Overexpressed rab11 mis-traffics into axons and is removed by predominantly retrograde transport. (**A,B**) Axon stem/proximal dendrite ratio (**A**) and axon stem signal intensity (**B**) of total rab11 (both transfected and endogenous) in cytosolic GFP or rab11a WT overexpressed neurons. Transfected neurons were immunofluorescently labelled with rab11 antibody and quantified. n = 3 independent staining, Welch's t-test. (**C**) Axon stem/ proximal dendrite immunofluorescence intensity ratio of transfected neurons. Neurons were transfected with cytosolic GFP or fluorescence-tagged rab11a forms, and the fluorescent protein was probed by immunofluorescence staining. (**D**) Comparison of fluorescence signal between proximal and distal axon. Distal axon intensity was divided by axon stem intensity. (**E**) Representative kymographs of fluorescence-tagged rab11a forms in axon stems and proximal dendrites. (**F**) Quantification of vesicular movements in neurons overexpressing fluorescence-tagged rab11a forms. Images were acquired every second for 3 min, and kymographs were produced from a 30 μm region of the axon stem or proximal dendrite. Vesicle movement direction (**F**) and average vesicle velocity (retrograde and anterograde *Figure 7—figure supplement 1A,B*) were quantified. Error bars represent s.e.m. rab11 overexpression results were accumulated from at least three independent transfected cultures. Sample numbers are described in the figure. (**B ~F**) One-way ANOVA followed by Bonferroni's post hoc test. Black asterisks: axon and dendrite comparison, blue asterisks: rab11 type comparison. *p<0.05, **p<0.01, ***p<0.001 and ****p<0.0001.

DOI: https://doi.org/10.7554/eLife.26956.016

The following figure supplement is available for figure 7:

**Figure supplement 1.** Quantification of vesicular movements in neurons overexpressing fluorescence-tagged rab11 constructs.

DOI: https://doi.org/10.7554/eLife.26956.017

## Overexpression of rab11 enhances regeneration

As shown above, newly plated neurons show robust rapid axon growth and a high proportion of them regenerate after axotomy. Since this matches the period when axonal rab11 is still abundant, we asked if axons could regain their regenerative state if rab11 was returned to the axon at 16 DIV. In principle this should enable transport of many growth-related molecules into the mature axon. Taking advantage of mis-trafficking after overexpression to re-introduce rab11 into axons, we axotomized axons of rab11 WT, DN and CA transfected neurons in the proximal zone. Staining for total rab11 in transfected neurons confirmed that transfection had increased total rab11 in the proximal axons, and the proximal axon/dendrite ratio of immunofluorescence intensity was increased (*Figure 7A,B*).

Axons were cut in the proximal region where rab11 was now present and their regeneration behaviour was measured. Laser axotomy led to an accumulation of fluorescent rab11 for all forms in the retraction bulb (*Figure 8A*, *Figure 8*, *Video 3*), showing that transfected rab11 is present during the reorganization of the severed stump. Interestingly, in some cases this accumulation started even before (~30 min) the reflux of the cytosolic GFP or bulb formation (*Figure 8A*, *Video 3*). We then analysed the regenerative measures as in the previous experiments. As with GFP-transfected neurons, retraction distance and time to form a retraction bulb was correlated (*Figure 8B*), and the retraction distance of rab11a transfected neurons showed a biphasic distribution for all forms (*Figure 8C*), so we again separated neurons into short and long retraction groups. The percentages of axons in the short and long retraction groups did not change between GFP and the other rab11 forms. But strikingly, for all rab11 forms, overexpression resulted in decreased retraction in the long retraction group (*Figure 8E*). The presence of rab11 also influenced growth cone regeneration, and the long retraction group of rab11 neurons had an improved regeneration ratio, but only with the WT and DN form suggesting the importance of the inactivated state of rab11 (GFP 11%, WT 38%, DN 38%, CA 13%, GFP vs WT p=0.0174, GFP vs DN p=0.0342, *Figure 8F*). No pro-regeneration effect was observed in the short retraction group (*Figure 8F*).

For other measures, regeneration initiation time was unchanged (*Figure 8G*), but the length of the axons after 2 hr was increased with rab11 WT and CA (*Figure 8H*). Growth cone area was also enlarged in regenerating neurons transfected with rab11 WT, and a trend was seen with the CA but did not reach significance (*Figure 8I*). In summary, increasing rab11 in the axons of 16 DIV neurons led to enhancement of regeneration especially in the long retraction group. The activation state of rab11 was a significant factor, with the DN form promoting growth cone formation and the CA form stimulating subsequent elongation.

## Overexpression of rab11 enables integrin transport into axons

Our hypothesis was that rab11 forced into axons by overexpression would carry with it molecules that promote axon regeneration. We therefore transfected neurons with rab11-GFP or control GFP at DIV10, the time at which partitioning of transport between axons and dendrites becomes apparent. At DIV 16 the neurons were stained for GFP and for a growth-associated receptor normally excluded from mature neurons. For this growth-related molecule normally carried in rab11 vesicles we selected α5 integrin (*Caswell and Norman, 2006*; *Eva et al., 2010*); *Gardiner et al., 2007*; *Hülsbusch et al., 2015*), which is present in neurons and can be detected with a reliable antibody. The level of α5 immunostaining was measured by drawing a linear AOI along the axons (identified by staining the AIS with neurofascin) and dendrites and subtracting the same AOI moved to the background. The results were plotted as axon/dendrite intensity ratio (*Figure 9B*) and absolute level (*Figure 9C*). As above, overexpression of rab11-GFP led to its presence in the axons over approx. the proximal 200 μm (*Figure 9A*). The distribution of α5 integrin changed similarly. In control GFP-transfected neurons α5 integrin was not detectable in axons (*Figure 9A–C*), while in rab11-GFP-transfected neurons the staining intensity was similar between proximal axons and dendrites (*Figure 9A–C*). This experiment shows that rab11 is able to carry growth-related receptors into axons.

## Axonal rab11 function is conserved in human neurons

Next, we investigated whether the relationships between maturation, rab11 distribution, axonal expression and regeneration are conserved in human neurons, taking advantage of recent advances that have made it possible to generate human dopaminergic neurons in vitro. The human embryonic stem cell (hESC) line RC17 (Roslin Cells) was induced to dopaminergic neuronal differentiation using a protocol in which differentiated post-mitotic neurons are generated within day (d)35 after neural induction (*Kirkeby et al., 2012*), and the cells were then allowed to mature in vitro for up to d65. As in the rodent neurons, laser-mediated axotomy revealed a maturation-related loss of capacity for axonal regeneration. Between d35-40, successful axonal regeneration was seen in over 70% of neurons, but this dropped to 46.9% and 39.6% between d45-55 and d55-65 respectively (*Figure 10A*).

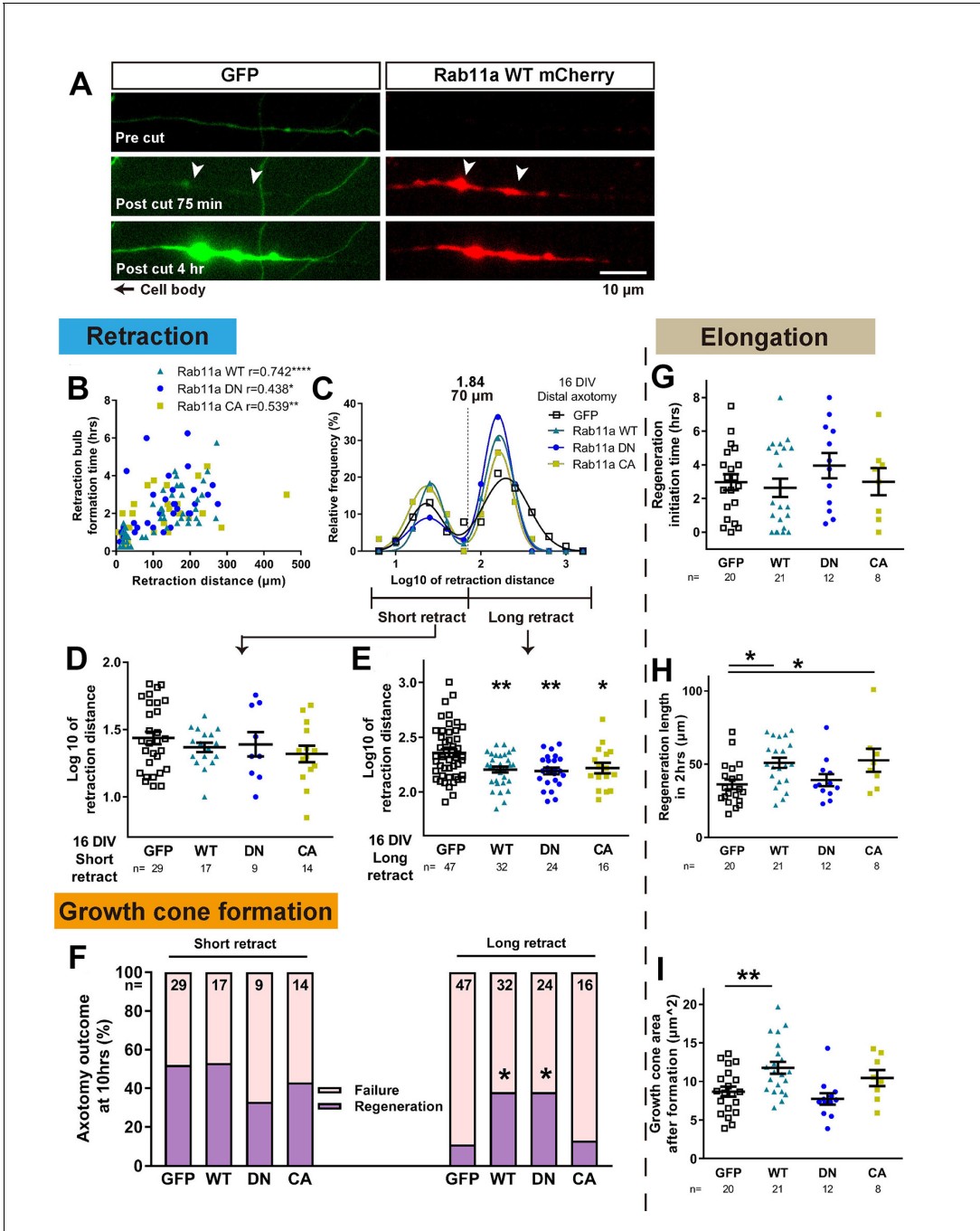

**Figure 8.** Rab11 overexpression increases intra-axonal rab11 and enhances regeneration. (**A**) Accumulation of overexpressed rab11 WT at the retraction bulb. Note how rab11a accumulation can start earlier than GFP. (**B,C**) Retraction distance and retraction bulb formation time correlate (**B**), and the frequency plot of retraction distance takes a biphasic distribution (**C**) in rab11 transfected neurons. (**D,E,F**) Retraction distance (**F,G**) and regeneration ratio (**H**) of rab11 overexpressed neurons categorized by retraction group and form. F, G: One-way ANOVA followed by Dunnett's post hoc test. H: Fisher's exact test with Bonferroni correction. (**G,H,I**) Regeneration initiation time, regeneration length and growth cone area of rab11 overexpressed neurons categorized by rab11 type. One-way ANOVA followed by Dunnett's post hoc test. Error bars represent s.e.m. Axotomy results were accumulated from at least three independent sessions. Retraction distance was converted by log10. Sample numbers are described in the figure. *p<0.05, **p<0.01 and ***p<0.001.

DOI: https://doi.org/10.7554/eLife.26956.018

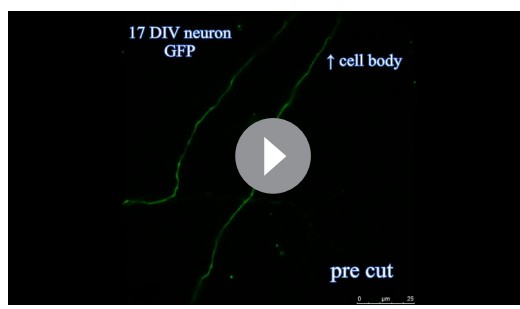

**Video 3.** shows an example of a neuron transfected with rab11 DN-RFP and GFP, showing that rab11 appears in the stump soon after axotomy and is present in the growth cone throughout regeneration and growth. Table
DOI: https://doi.org/10.7554/eLife.26956.019

To determine whether this maturation-related loss of regenerative capacity was associated with a reduction in rab11 trafficking into the axon, we expressed rab11-GFP in the human neurons. At d41, the level of rab11-GFP fluorescence was similar in axons and dendrites. At d51 and d65 rab11-GFP was just detectable in the axons but there was a significant drop relative to levels in the dendrites (*Figure 10B, C*). We asked whether, as in the rodent neurons, rab11 overexpression could restore the loss of axonal regeneration potential. Transfection of rab11 did not affect the rate of axonal regeneration in d35-40 neurons, but a significant increase in the number of axons showing regeneration was seen in d45-55 axons (*Figure 10A*) with an associated increase in the extent of elongation at 2 hr post-axotomy (*Figure 7D*) These results confirm that, as in rodent neurons, there is an age-dependent loss of axonal regeneration in human CNS neurons that is associated with loss of axonal rab11.

## Discussion

In order to analyse why CNS neurons lose their ability to regenerate with maturity, and to develop regeneration-promoting treatments, a manipulable in vitro model is needed. We describe an in-vitro single neuron axotomy model, in which there is a progressive decline in the intrinsic regeneration

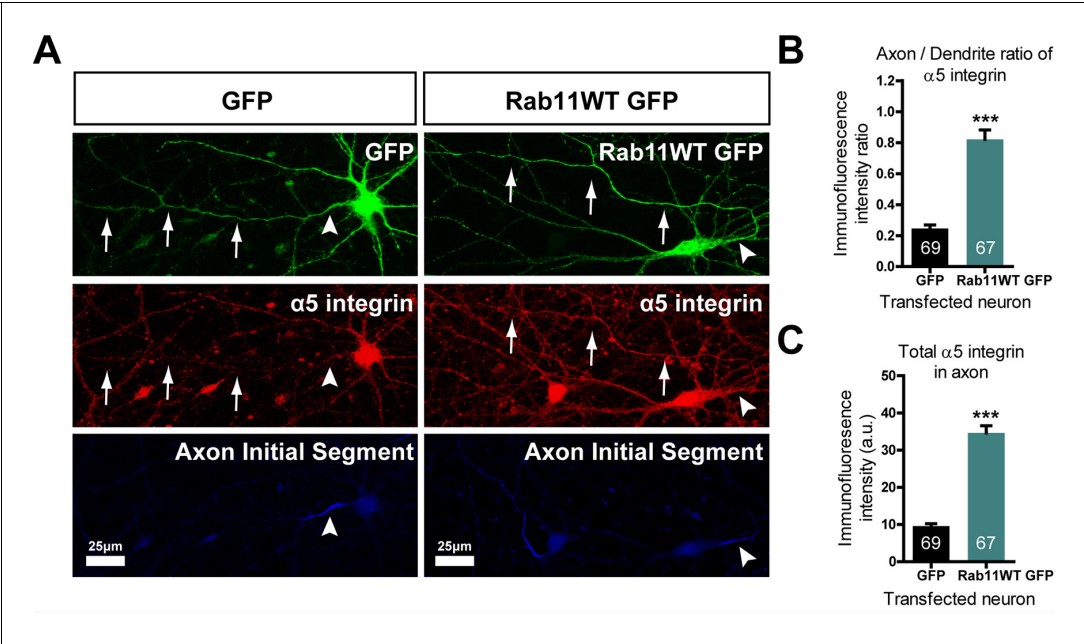

**Figure 9.** Rab11 forced into axons by overexpression carries α5 integrin with it. DIV 10 neurons were transfected with either rab11-GFP or control GFP, then fixed and immunolabelled on DIV16. (**A**) left shows that in control cells GFP enters dendrites and the axon identified by neurofascin staining, while α5 integrin is excluded. On the right, rab11-GFP is transfected, and mistrafficks into the proximal axon. The axon now contains plentiful α5 integrin. The distribution of α5 integrin between axons and dendrites is quantified in (**B**) and (**C**) as fluorescence intensity levels and axon/dendrite intensity ratio. Bar = 25 μm, ***=P < 001 by student's t test.
DOI: https://doi.org/10.7554/eLife.26956.020

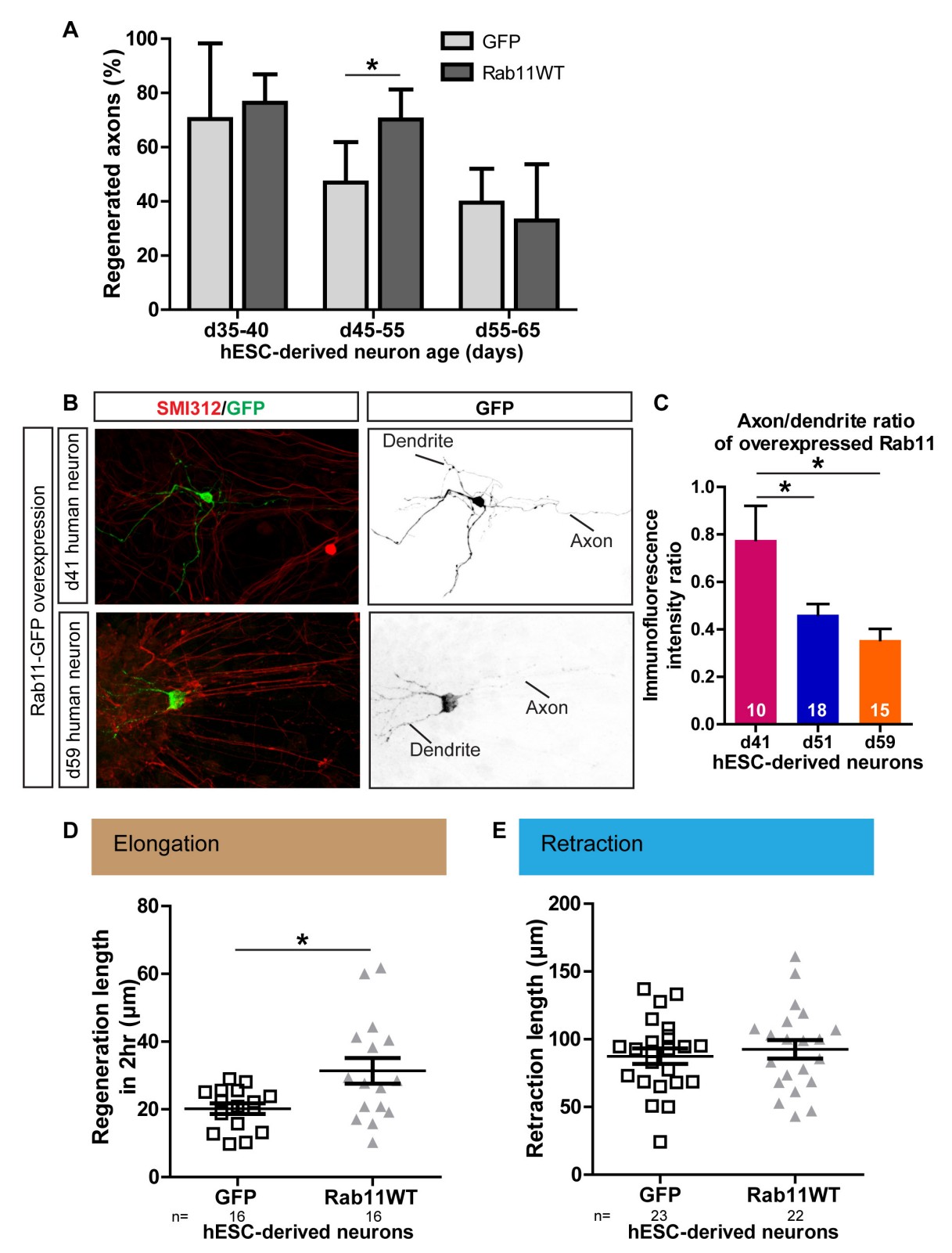

**Figure 10.** Age-dependent decline in axonal regeneration of hESC-derived neurons is partially rescued by increasing intra-axonal rab11 levels. (**A**) hESC-derived neurons show an age-related decrease in regeneration following laser-mediated axotomy. Overexpression of rab11-WT improves the percentage of regenerated axons in d45-55 neurons. n = 4 independent experiments per time point. (**B,C**) Axonal levels of rab11-GFP decline in mature hESC-derived neurons as compared to dendritic levels. Axons were identified using SMI-312 immunolabeling and intensity of GFP immunolabeling was

*Figure 10 continued on next page*

*Figure 10 continued*

measured. n > 3 independent stainings. (**D**) Regeneration length measured at 2 hr after initiation of regeneration is increased in rab11 overexpressing hESC-derived neurons. n = 16 for both for GFP and rab11 collected from four independent experiments. (**E**) Retraction distance distribution is not affected by rab11 overexpression in hESC-derived neurons. n = 22 and 23 for GFP and rab11 +GFP respectively, collected from four independent experiments. In all graphs, error bars represent S.E.M. *p<0.05. Unpaired t-test, Welch's correction.

DOI: https://doi.org/10.7554/eLife.26956.021

ability of the axons of cortical neurons as they mature. The neurons in these cultures can be transfected and regeneration-promoting effects of pharmaceuticals and other interventions tested (unpublished observations). We describe here an analysis which demonstrates that the development of polarised neuronal transport is an important factor in the loss of regeneration ability, through exclusion of rab11 recycling endosomes (which contain key growth molecules) from axons. The main influence on regeneration in this study was neuronal maturity, represented by days in vitro (DIV). This is in line with many in vivo studies which show that the regenerative ability of CNS axons is lost with maturity. Neuronal maturity is also a key factor in vivo, where grafted embryonic neurons have shown extensive growth in the adult brain and spinal cord, yet mature neurons show no growth (*Jakeman and Reier, 1991*; *Kim et al., 2006*)(*Lu et al., 2012*). In *C.elegans* axons also lose regenerative ability with age, although the mechanism may be rather different (*Tang and Chisholm, 2016*; *Byrne et al., 2014*). Even when regeneration is stimulated by manipulation of signalling pathways using Phosphatase and Tensin Homolog Deleted from Chromosome 10 (PTEN) deletion, the knockdown is much more effective if it is performed during the growth phase of cortical neurons rather than adulthood (*Du et al., 2015*; *Geoffroy et al., 2016*). However some regenerative events are seen after CNS axotomy in adulthood. In the corticospinal tract and other pathways extensive lateral sprouting can occur (*Bareyre et al., 2004*), particularly in primates (*Rosenzweig et al., 2010*), and growth into embryonic grafts can occur (*Bernstein-Goral and Bregman, 1993*; *Kadoya et al., 2016*).

Detailed quantification of the events following axotomy revealed behaviours that change at different DIV. The retraction distance after axotomy increased with maturity; at 4 DIV retraction distances were short, but by 24 DIV they were much longer, with a mixture of short and long retractors at 16DIV. Retraction distance predicted regeneration with long retractors seldom regenerating, especially after distal axotomy. The correlation between long retraction and regeneration failure has also been shown in vivo after cutting intracortical axons by laser (*Canty et al., 2013*). We suggest that the increased retraction distance with maturity may reflect changes in cytoskeletal dynamics together with decreased anterograde transport of materials required to maintain the axons. At DIV16, where we see both long and short-retracting axons indicating a mixed degree of maturity in neurons as they are becoming polarized and selective transport is being established (*Franssen et al., 2015*). By DIV 24 these changes are fully established and all axons show long retraction after axotomy and almost complete loss of regenerative ability. The speed of axon growth after initiation of regeneration and the size of regenerated growth cones declined earlier than these retraction changes, the changes being complete by 16 DIV. Growth cone size is affected by many environmental and intrinsic factors, does not correlate with axon growth speed (*Harris et al., 1987*; *Hur et al., 2012*; *Tosney and Landmesser, 1985*), and is probably not a useful general indicator of regenerative ability.

We focused on the hypothesis that the developmental change in regenerative ability in axons is related to the progressive exclusion of growth-related molecules from axons (*Bentley and Banker, 2016*; *Britt et al., 2016*). In particular, previous work has shown that integrins (*Franssen et al., 2015*) and Trks (*Hollis et al., 2009a*) become excluded from axons after they mature, as are most postsynaptic proteins. Precise control of the location of intracellular molecules is necessary so that axons or dendrites can have different structures and functions. Many growth-related molecules are transported into axons via the recycling pathway in vesicles marked by the small GTPase rab11 (*Lasiecka and Winckler, 2011*)(*Baetz and Goldenring, 2013*; *Welz et al., 2014*). In this study, we show that rab11 is present in immature axons but becomes restricted to a somatodendritic distribution with maturity, correlating with the loss of ability to regenerate axons in both rodent cortical neurons and human dopaminergic neurons. As with many molecules the selective transport mechanism is somewhat leaky, which could indicate passive diffusion of vesicles and molecules into axons

followed by selective removal by retrograde transport. Overexpression of rab11 leads to mis-trafficking and therefore leads to the presence of some rab11 in the proximal axons of mature neurons. This allowed us to ask whether the presence of rab11 affects regeneration.

In our study, rab11 WT and CA enhanced regeneration length and growth cone size, and this is in line with previous studies where these molecules have been shown to be involved in axon and dendritic growth (*Park et al., 2006*). Our results also demonstrated that along with WT the DN mutant enhances formation of a new growth cone to initiate regeneration whereas the CA does not. Other studies have also confirmed the protrusion-initiating properties of rab11 DN (*Shirane and Nakayama, 2006*; *Ramel et al., 2013*), suggesting that the different forms of rab11 can have different effects on different processes including vesicle transport, trafficking and actin reorganization and can therefore affect different phases of regeneration. Rab11 overexpression also enhanced regeneration in human dopaminergic neurons. The regeneration-promoting effect of rab11 overexpression is assumed to be due to the transport into axons of the many growth-related molecules that are transported in this class of endosome (*Welz et al., 2014*). As an example, we showed that α5 integrin is transported into the proximal axons of rab11 overexpressing neurons.

Rab11 transport is controlled in several ways. A transport complex of rab11, Arf6 and JIP3/4 can associate with kinesin or dynein depending on the activation state of Arf6 (*Montagnac et al., 2009*) and rab11 associates with kinesin through activated protrudin (*Matsuzaki et al., 2011*). Enhancing the level of PIP3 through PTEN knockdown has been a successful method for inducing regeneration (*Liu et al., 2010*; *Du et al., 2015*). Our results fit with this story, because PIP3 has effects on transport through GAPs and GEFs and on membrane trafficking (*Macia et al., 2008*; *Randazzo et al., 2001*). It is probable that there is a cycle, where transport of receptors allows PI3K activation and PIP3 generation, which in turn enhances transport and trafficking (*Cheng et al., 2011*). Manipulation of rab11 transport using these methods can enhance regeneration in mature cortical neurons (unpublished observations), and will be the basis of future in vivo investigations.

The changes in transport that we have shown affect regeneration, but CNS axons nevertheless show some regenerative activity after damage, particularly lateral sprouting and growth into embryonic tissues. It is unlikely that these events can happen without the presence of a receptor on the axons and a corresponding ligand in the environment. These receptors, currently unidentified, must continue to be transported into mature axons, presumably independently of rab11.

Overall this study supports the concept that CNS axons lose their intrinsic ability to regenerate with maturation. Mature axons lose regeneration-associated molecules due to the development of polarised transportation, which directs the many growth-related molecules in rab11 vesicles to dendrites and away from axons. Modifying selective axon transportation could therefore be a strategy for enhancing intrinsic regeneration ability in the adult CNS.

## Materials and methods

### Experimental procedures

#### Cell culture

Cortical neurons were derived from E18 Sprague Dawley rat cortex, and were plated at the density of 100–150 cells/mm$^2$. The medium used for plating was a combination of fresh B27 medium (neurobasal medium supplemented with 2% B27% and 1% Glutamax) and B27 medium conditioned by astrocytes.

Astrocytes were derived from P0-2 Sprague Dawley rat cortex, and were grown for 7–8 days before being frozen for future use. They were then thawed and further grown for another 5–6 days before being combined with neurons as co-culture, or used for B27 medium conditioning. The co-culture method was adapted from a previously established method (*Kaech and Banker, 2006*).

#### Plasmid transfection

Neurons were transfected via microporation or magneting transfection. Microporation was done by using the Neon Transfection System (ThermoFisher (Waltham, Massachusetts, United States)), following the manufacturer's protocol. Neurons were plated at a higher density of 200–300 cells/mm$^2$ to compensate for cell death. This method was mainly used for axotomizing at 4 and 24 DIV.

The Magnefect System (NeuroMag OZ Biosciences, France) was mainly used on neurons cultured for 13–14 DIV, following the manufacturer's protocol. DNA transfected with this method expressed within 2 DIV, and was mainly used for 16 DIV axotomy experiments which required co-transfection of plasmids.

Cytoplasmic GFP was from Lonza (Switzerland) (pmax GFP), and rab11a WT GFP/mCherry has been described previously (*Eva et al., 2010*). Rab11a DN RFP/CA RFP were kindly provided by Dr. Gonzalo Solis (Université de Lausanne).

## Live neurofascin staining

Pan neurofascin extracellular antibody was dialyzed overnight by a 12–14 kDa dialyzer (D-Tube Dialyzer Maxi, MWCO 12–14 kDa/ Millipore, Billerica, Massachusetts) against 1xPBS (594 nm wavelength) or 0.1M borate buffer (650 nm wavelength). Then the dialyzed antibody was labeled with a 594 (650) nm wave length dye by following the protocol of the DyLightTM 594 Microscale Antibody Labeling Kit (ThermoFisher). The labeled antibody was added to the culture immediately before the axotomy session at 1:1000, and was not removed.

## Laser axotomy

The laser chamber was kept at 37 degrees at all times, and CO2 was set at 10%. The immunofluorescence exposure was minimized in order to control phototoxicity. Axotomy was performed by an UV Laser (355 nm, DPSL-355/14, Rapp OptoElectronic, Germany) connected to a Leica DMI6000B (Leica Systems, Germany), and all images were taken with an EMCCD camera (C9100-02, Hamamatsu).

Neurons transfected with cytosolic GFP were axotomized at different DIV (4, 16, 24 DIV) and distance (250–2000 µm) from the cell body. Large neurons with the anatomy of projection neurons, with several dendrites and a single axon were selected. All axons were confirmed by checking the live neurofascin staining, and neurons with more than two axons were excluded from the study. Astrocytes were not removed during axotomy.

Axotomy distance was defined as the axon length between the stem of the axon and the laser shot epicenter, which was manually traced and measured using the Leica application suite AF. To minimize the influence of branch points and adjacent branches, axotomy locations were chosen at least beyond the first branch point, and as far away as possible from any visible branch points. All efforts were made to keep the laser power consistent and cutting to a minimum, but priority was given to make a clean cut.

Post laser response was imaged every 20–30 min for every axotomized neuron, starting immediately after axotomy. Imaging was continued in many cases more than 15 hr, and the first 10 hr were used for analysis. Axotomy results were pooled into one group, but axotomy sessions which suffered large amounts of death were excluded. All axotomy results were manually analysed using the Leica application suite AF software.

## Live imaging for vesicle tracking

All live-cell imaging for rab11 vesicle movement studies was performed using an Olympus IX70 microscope equipped with a CCD camera (ORCA-ER, Hammamatsu, Japan) and a PerkinElmer UltraVIEW scanner for spinning disk confocal microscopy. MetaMorph software was used to operate the system.

Rab11 vesicle movements were imaged at the proximal axon and dendrite; the first 100 µm segment was considered as proximal. Images were taken one frame/second for 3 min. Kymographs were obtained from the most well focused 30 µm strip of the axon or dendrite. Vesicles moving only in one direction for more than 2 µm were defined as either anterograde (towards the distal axon) or retrograde (towards the cell body). Vesicles moving in both directions over 2 µm was termed as bidirectional. The average speed for every anterograde or retrograde movement during the imaging period was quantified, and their average was reported as the cell's anterograde or retrograde vesicle velocity. All analysis was performed with FIJI.

## RNA sequencing

Total RNA was extracted from 1, 4, 8, 16, and 24 DIV cultures by using the Qiagen RNeasy Micro Kit (Qiagen, 74004), following the manufacturer's instruction. six samples were used for each time point.

The extracted RNA was quality checked with the Agilent Bioanalyzer Pico Chips (RIN > 7.0) and 2 ng was used for whole transcriptome amplification using NuGEN RNAseq System V2 to generate ds-cDNA. Subsequently, 2 ug of amplified cDNA was subjected to library preperation using NuGEN Ovation Multiplex System. This was then sequenced using an Illumina HiSEQ 2500 instrument, which yielded an average of 12.35M single-end 50 bp reads per sample. The reads were then aligned onto the Rattus Norvegicus Rnor 5.0 genome using Tophat (v2.0.8, RRID:SCR_013035) with a mapping rate of 91.67%, and used for differential expression analysis using Cufflinks (v.2.2.1 RRID:SCR_014597). Genes were defined as increasing (decreasing) if (1) the highest FPKM value was more than 3 folds of the lowest FPKM, (2) the false discovery rate adjusted p value between the highest and lowest FPKM calculated by Welch's t-test was smaller than 0.05, and (3) at least one of the time points marked a FPKM value of more than 10. Increasing and decreasing genes were subjected to pathway analysis using the Ingenuity pathway analysis program.

## Patch clamping

Whole-cell patch-clamp recordings were performed on 4, 8, 16, 24 DIV neurons, and the neurons' passive membrane properties and spike properties were measured. 800 ms long negative and positive currents were applied in an alternating manner, and the currents were increased in 20 pA steps. The results were recorded in current-clamp mode. All spike events that accelerated over 20 mV/ms$^2$ and reached 0 mV were classified as APs.

The resting membrane potential was measured immediately after patching the cell and before applying any current. The input resistance was defined as the change in voltage caused by an injection of a hyperpolarizing current, divided by the amount of current. The neuron response to one of the first three negative steps of the alternating currents was used for calculation.

For measuring spike properties, the response elicited by the 200 pA current was used. Spike frequency was measured by dividing the number of action potentials by the current injection time. Spike threshold was defined as the difference between the resting membrane potential and the neuron's membrane potential when spike acceleration went over 20 mV/ms$^2$ for the first time. Spike amplitude was measured as the difference between the membrane potential at the threshold point and at the peak of the spike. The spike width was defined as the time it took from the spike upstroke's half-maximal amplitude to the same voltage during the repolarizing phase. Finally, spontaneous activity was considered as positive if the patched neuron produced any action potentials within two minutes without any current injection.

## Immunocytochemistry

Neuron cultures were fixed for 12 min at room temperature by using 3.6% PFA pH 7.4 with 3% sucrose. Fixed cultures were blocked with blocking buffer (5% FBS and 0.1% Triton X-100 in 1xPBS) for 60 min. Then they were probed with primary antibodies overnight at four degrees. Secondary antibodies were applied for 1 hr at room temperature. For imaging, Leica DMI4000B microscope (Leica Systems) with Leica TCS SPE confocal system for laser scanning and detection was used. Care was taken to keep the images in linear range. Leica application suite AF software was used for immunofluorescence intensity quantification. In all cases where confocal microscopy was used, image stacks were converted to maximum intensity projection images. All immunofluorescence intensities were calculated by measuring the intensity of the region of interest (ROI), and then subtracting the intensity of a control region adjacent to the ROI to compensate for non-specific staining. Nearby prominent structures were avoided for control regions.

The immunofluorescence intensity of the 0–100 μm region of the axon was considered as the 'axon stem signal'. For dendrites, the brightest three branches were chosen, and the 'dendrite signal' was calculated by averaging the immunofluorescence intensities of the 0–100 μm region of these three dendrites. Cell body signal was measured at the soma, by cutting off the dendrite and axon area from their stems. For '400 μm axon signal' and '800 μm axon signal', immunofluorescence intensities of axon regions 350–450 μm and 750–850 μm away from the cell body were measured.

## Antibodies

Primary antibodies: VGLUT1 (Synaptic systems, 135–303, RRID:AB_887875), VGAT (Synaptic systems, 131–003, RRID:AB_887869), PSD95 (ThermoFisher, MA1-04, RRID:AB_325399), Gephyrin (Synaptic systems, 147–011, RRID:AB_887717), extracellular Pan-Neurofascin (NeuroMab, 75–172, RRID:AB_2282826), cleaved caspase 3 (Cell signaling, 9664S, RRID:AB_2070042), rab11 (ThermoFisher, 71–5300, RRID:AB_2533987), β3 Tubulin (abcam, ab41489, RRID:AB_727049), SMI312 (abcam, ab24574), Ankyrin G (NeuroMab, 75–146), GFP (ThermoFisher, A10262), RFP (abcam, ab62341, RRID:AB_945213), integrin alpha5 (MERCK, AB1928, RRID:AB_2128185).Secondary antibodies: Alexa Fluor 488 goat anti chicken, 568 goat anti rabbit, 660 goat anti mouse (ThermoFisher).

## Culture of hESC-derived neurons

RC17 clinical grade hESC cells (RRID:CVCL_L206) were sourced from Roslin Cells, Scottish Centre for Regenerative Medicine, Edinburgh, United Kingdom. The cell line was genotyped and tested for genetic modifications by Roslin Cells. The cell line is free from mycoplasma contamination as determined by RT-qPCR. Complete details are published (*De Sousa et al., 2016*) and available online: https://hpscreg.eu/cell-line/RCe021-A

On d0 hESC were detached with EDTA and transferred to form embryoid bodies from d0 to d4 in neural induction medium (Neurobasal:DMEM/F12 (1:1, ThermoFisher Scientific), 0.2% P/S, l-glutamine, N2 (Gibco, Life Technologies), B27 (ThermoFisher Scientific), supplemented with recombinant Sonic Hedgehog C24II (Shh, 200 µg/ml, Minneapolis, MN), recombinant noggin (Ng, 100 µg/ml, R and D systems) SB431542 (SB, 10 mM, R and D systems) and CHIR99021 (CH, 10 mM, Tocris Bioscience, UK). Rock inhibitor (RI, 10 MM, Sigma-Aldrich, St. Louis, Missouri) was present in the medium from d0 to d2. At d4, embryoid bodies were plated in poly-L-ornithine (0.001%, Sigma-Aldrich), Laminin (4.4 µg/ml, ThermoFisher Scientific) and Fibronectin (PLF) coated plates and cultured in neural proliferation medium (Neurobasal:DMEM/F12 (1:1), 0.5xN2, 0.5xB27, supplemented with Shh, Ng, SB and CH from d4-d7, and with Shh, Ng and CH from d7-d9). At d11, cells were dissociated with Accutase (Sigma-Aldrich) and $50*10^4$ cells per well single cell suspensions were plated in PLF-coated 8-well chamber slides (Ibidi). Cells were cultured in neuronal differentiation medium (NB with 0.2% P/S, l-glutamine, B27, Ascorbic Acid (200 mM, Sigma-Aldrich), recombinant human Brain-derived Neurotrophic Factor (BDNF, 20 µg/ml, R and D systems), Glial-cell line derived Neurotrophic Factor (GDNF, 100 µg/ml, R and D systems) which was further supplemented with Dibutyryladenosine 3', 5'-cyclic monophosphate sodium salt (db-CAMP; 50 mM, Sigma-Aldrich) and DAPT (10 mM, TOCRIS) from d14 onwards. Medium was replaced twice weekly up to day 50, after which medium was replaced weekly.

## Plasmid transfection

Human neurons were transfected using a modified Lipofectamine 2000 protocol as previously published (*van Erp et al., 2015*). Per 8-well chamber slide, 1.8 ug plasmid DNA was mixed with 3 µl of Lipofectamine 2000 in 200 µl of NB, incubated for 30 min, and then added to the neurons in transfection medium (NB with l-glutamine and B27) at 37°C in 5% CO2 for 45 min. Next, neurons were washed with NB, the original conditioned medium was replaced, and cells were cultured for 2 days at 37°C in 5% CO2.

## Laser-mediated axotomy of hESC-derived neurons

Selection of neurons and conditions of axotomy were as described for rodent neurons. An Andor spinning disk confocal microscope with temperature and CO2 controlled live imaging chamber was used. GFP-transfected neurons were selected and the axon was identified based on morphology. For axotomy, a MicroPoint photo-stimulation tool with 365 nm laser in combination with a 40xH2O objective were used. The distance from the cell body was kept similar for all experiments (between 300–500 um). Both axon and cell body were imaged for 16 hr following axotomy with 20 min intervals using a 20x objective and minimal laser intensity to reduce the risk of phototoxicity. Neuronal survival was high (typically 100%). Axotomy results were manually analysed using Fiji.

## Immunocytochemistry

For human neurons, cells were fixed by addition of 8%PFA to equal volume of culture medium (final concentration 4%) and incubating for 15 min at RT. Then cultures were blocked for 30 min at RT in blocking buffer (2% Normal Horse Serum, 2% Normal Goat Serum, 1% bovine serum albumin (BSA), 0.2% Triton X-100) followed by overnight incubation with primary antibodies in blocking buffer. The following day, samples were incubated with secondary antibodies diluted in blocking buffer for 60 min at RT. Images were taken using a Leica TCS SP8 confocal microscope with 40x objective. Axons were identified using *SMI312 (*Biolegend, San Diego, California).

## Quantification of integrin alpha5 axon-dendrite ratio

Cortical neurons were transfected at DIV10 with either GFP or rab11WT and fixed at DIV14 with methanol. These were immunolabelled for integrin alpha5 and GFP. Control and rab11 transfected cultures were fixed and labelled together, using identical conditions. Images were acquired by confocal laser scanning microscopy using a Leica TCS SPE confocal microscope, using identical settings for each image. Images were acquired at 40x to include the cell body, dendrites and the proximal section of axon in each image. Leica LAS AF software was used to measure mean fluorescnece in the axon, and in two dendrites (to give a mean dendrite measurement). A region next to each neurite was used to subtract background fluorescence. Axon dendrite-ratio was determined as the mean dendrite fluorescence intensity divided by the axon intensity. GraphPad Prism (RRID:SCR_002798) was used for statistical analysis of data using a students T-Test.

## Statistics

Statistics were performed using GraphPad Prism and the program R. Kolmogorov–Smirnov test was used for normality testing. For comparison of 2 sample groups, t-test was used. Student's or Welch's t-test was used depending on the variance. For comparison of more than two groups, one-way ANOVA was normally used. For the post hoc test, Bonferroni's, or Dunnett's post hoc test was used depending on the comparisons that were going to be made. For the cases where the assumption of equal variance was significantly violated (determined by Bartlett's test), Games-Howell post hoc test was used. Some of the patch clamp results and many of the axotomy results violated the normal distribution assumption. For the patch clamp results, Kruskal-Wallis test was used instead of one-way ANOVA. For the axotomy results, the data was converted by log10 to shift it to normal distribution. For regeneration ratio results, fisher's exact test was used, and Bonferroni correction was applied for repeated comparisons. For the rab11 vesicle movement analysis, two-way ANOVA was used, followed by a Bonferroni's post hoc test.

## Acknowledgements

The work was supported by grants from the Glaxo Smith Kline International Scholarship (to HK), Honjo International Scholarship (to HK), Bristol-Myers Squibb Graduate Studentship (to HK), the Christopher and Dana Reeve Foundation, the Medical Research Council, the ERC advanced grant ECMneuro, the NIHR Cambridge Biomedical Research Centre and the Operational Programme Research, Development and Education in the framework of the project 'Centre of Reconstructive Neuroscience', registration number CZ.02.1.01/0.0./0.0/15_003/0000419

## Additional information

### Funding

| Funder | Grant reference number | Author |
| --- | --- | --- |
| GlaxoSmithKline | International Scholarship | Hiroaki Koseki |
| Honjo International Scholarship Foundation | | Hiroaki Koseki |
| Bristol-Myers Squibb | Graduate Studentship | Hiroaki Koseki |
| European Molecular Biology Organization | Long Term EMBO Fellowship (ALTF 1436-2015) | Susan van Erp |

| Ministerstvo Školství, Mládeže a T?lovýchovy | CZ.02.1.01/0.0./0.0/15_003/0000419 | Jessica CF Kwok James W Fawcett |
| --- | --- | --- |
| Medical Research Council | G1000864 | James W Fawcett |
| Christopher and Dana Reeve Foundation | International Consortium | James W Fawcett |
| European Research Council | ECMneuro | James W Fawcett |
| National Institutes of Health | Cambridge Biomedical Research Centre | James W Fawcett |

The funders had no role in study design, data collection and interpretation, or the decision to submit the work for publication.

## Author contributions

Hiroaki Koseki, Conceptualization, Formal analysis, Validation, Investigation, Visualization, Methodology, Writing—original draft, Writing—review and editing; Matteo Donegá, Veselina Petrova, Susan van Erp, Investigation, Methodology; Brian YH Lam, Data curation, Investigation, Methodology; Giles SH Yeo, Conceptualization, Data curation, Supervision; Jessica CF Kwok, Richard Eva, Investigation, Methodology, Project administration, Writing—review and editing; Charles ffrench-Constant, Conceptualization, Funding acquisition, Project administration; James W Fawcett, Conceptualization, Formal analysis, Supervision, Funding acquisition, Writing—original draft, Project administration, Writing—review and editing

## Author ORCIDs

Susan van Erp ⓘ http://orcid.org/0000-0003-0883-2795
Richard Eva ⓘ http://orcid.org/0000-0003-0305-0452
James W Fawcett ⓘ http://orcid.org/0000-0002-7990-4568

## Decision letter and Author response

Decision letter https://doi.org/10.7554/eLife.26956.025
Author response https://doi.org/10.7554/eLife.26956.026

## Additional files

### Major datasets

The following dataset was generated:

| Author(s) | Year | Dataset title | Dataset URL | Database, license, and accessibility information |
| --- | --- | --- | --- | --- |
| Lam B, Koseki H | 2016 | Gene expression profiling of cultured embryonic rat cortical neurons | https://www.ncbi.nlm.nih.gov/geo/query/acc.cgi?acc=GSE92856 | Publicly available at NCBI Gene Expression Omnibus (accession no: GSE92856) |

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
