## [Decision Letter]

Thank you for submitting your article "Selective rab11 transport and the intrinsic regenerative ability of CNS axons" for consideration by *eLife*. Your article has been reviewed by three peer reviewers, and the evaluation has been overseen by a Reviewing Editor and K VijayRaghavan as the Senior Editor. The following individuals involved in review of your submission have agreed to reveal their identity: Mark H Tuszynski (Reviewer #1); Fengquan Zhou (Reviewer #3).

The reviewers have discussed the reviews with one another and the Reviewing Editor has drafted this decision to help you prepare a revised submission.

Summary:

Koseki and colleagues report a new and fundamental mechanism that is associated with the failure of adult axonal regeneration: the transport of rab11 vesicles, which carry cargo necessary to support regeneration.

This is a novel and interesting set of studies. While in general it has been previously noted that axonal transport is enhanced in peripheral axons after injury and that central axons may lack this response to injury, this work takes these previous (poorly documented) claims to mechanistic depth, demonstrating a novel mechanism related to axonal transport that is lost as a function of neuronal maturation. They also that demonstrate that "correction" of altered axonal transport by over-expression of rab11 enhanced regeneration. This is a nicely conceived, executed and supported set of studies. The novelty is high, and of interest to a broad readership. It would be nice to have in vivo confirmation of the relevance of this mechanism, but that is not essential as the mechanistic findings are so nicely and elegantly provided.

On a general level some concerns arose which can be readily addressed in re-writing. Some figures are a bit hard to follow. Some observations are interesting but it would be really interesting to provide more mechanistic insights, e.g., the biphasic response, short vs long retracting, recycling endosomes. Some parts need more clarification, e.g., distal vs proximal axotomy (rationale, experimental model, etc). In parts the manuscript feels a bit diffuse and may be much improved by focusing on the most important points and presenting the most relevant data that support the hypothesis.

Specific suggestions for modifications and clarification follow.

Essential revisions:

1) The authors state: "These results suggest that the decline of intrinsic axon regenerative ability is due to selective exclusion of key molecules, and that manipulation of transport can restore regeneration," These statements may be a bit bold. More likely, the authors have identified a *contributing* mechanism to central axonal growth failure, and manipulation of transport can significantly improve (not necessarily "restore") regeneration. There are numerous other mechanisms limiting regeneration that also contribute to central regeneration failure. Along these lines, the authors argue that CNS axons such as corticospinal axons lose the ability to regenerate even in a permissive environment, such as a peripheral nerve graft. A recent report in Nat Med 2016 (Kadoya) showed that adult corticospinal axons do indeed regenerate into a permissive environment of a neural stem cell graft. This undermines the authors' argument. This by no means negates the authors' findings or the importance of their work, but perhaps they should present their findings in a modified context: overcoming maturation-associated alterations in axonal transport can enhance central axonal regeneration.

2) The authors might comment on in vivo extension of their work. Did the authors already attempt to extent these tools to in vivo models and encountered some limitations? In my view, this would not at all undermine the importance of this work or its suitability for publication. I also appreciate that in vivo experiments can fail for many reasons unrelated to the accuracy of the proposed mechanism, and that practical limitations in vivo may preclude clear documentation of an in vitro finding that is well documented, as in the case of the present results. The authors might at least discuss in vivo extension of their work, and issues that might be encountered.

3) The authors are using y-axes on a log scale in some of the later figures, likely to deal with a large spread in the data. This hinders however the ability to compare data presented in other panels on a linear scale. Please reconsider whether a log10 scale is the only way to present these data.

4) Although the study showed that over-expression of rab11a in DIV 16 neurons significantly enhanced some of their regenerative responses to axotomy, such as decreased long retract and increased regeneration length in two hours, it is unclear how rab11 achieved such promoting effects. For instance, it is not clear why rab11a had different trafficking directions and velocities in axons and dendrites. It is also not clear why rab11a overexpression-induced mis-trafficking into axons enhanced some regenerative responses to axotomy. Can the authors show that over-expression of rab11a results in increased transport of axon growth-related molecules into the axon? Conversely, will the known axon regeneration promoting approaches affect the rab11 trafficking or localization in aged neurons? The answers to these questions will greatly strengthen the proposed hypothesis.

---

## [Author Response]

Summary:This is a novel and interesting set of studies. While in general it has been previously noted that axonal transport is enhanced in peripheral axons after injury and that central axons may lack this response to injury, this work takes these previous (poorly documented) claims to mechanistic depth, demonstrating a novel mechanism related to axonal transport that is lost as a function of neuronal maturation. They also that demonstrate that "correction" of altered axonal transport by over-expression of rab11 enhanced regeneration. This is a nicely conceived, executed and supported set of studies. The novelty is high, and of interest to a broad readership. It would be nice to have in vivo confirmation of the relevance of this mechanism, but that is not essential as the mechanistic findings are so nicely and elegantly provided.On a general level some concerns arose which can be readily addressed in re-writing. Some figures are a bit hard to follow. Some observations are interesting but it would be really interesting to provide more mechanistic insights, e.g., the biphasic response, short vs long retracting, recycling endosomes. Some parts need more clarification, e.g., distal vs proximal axotomy (rationale, experimental model, etc). In parts the manuscript feels a bit diffuse and may be much improved by focusing on the most important points and presenting the most relevant data that support the hypothesis.

Thank you, these are useful suggestions. The detailed response to individual points is below. We kept the Introduction and Discussion short, but we have now added some more mechanistic information and speculation. We apologise for the way the illustrations were presented in the constructed pdf, which led to a diffuse feel. The figures were not numbered in the review pdf, and it was not clear which figures were supplementary. *ELife* encourages supplementary figures attached to main figures, which we have done. In the resubmission we have put figure numbers underneath the figures to make things clearer. We worked hard on the figures, and we don’t think we can do better. For instance in Figure 4, which we admit shows complex information, we have divided the figure into panels, used coloured labels to guide the reader, and inserted arrows between the graphs to indicate the groups of data from short and long retractors that are being analysed. We do not want to take out details of axonal behaviour etc. from the paper, because we hope that other groups will want to use the assay we describe, and they will find these details useful. We have inserted information and speculation on short vs. long retraction and mechanism in several places in the discussion. The additional sections are highlighted in red, and we have not pasted them all here in the interests of compactness.

Essential revisions:1) The authors state: "These results suggest that the decline of intrinsic axon regenerative ability is due to selective exclusion of key molecules, and that manipulation of transport can restore regeneration," These statements may be a bit bold. More likely, the authors have identified a *contributing* mechanism to central axonal growth failure, and manipulation of transport can significantly improve (not necessarily "restore") regeneration. There are numerous other mechanisms limiting regeneration that also contribute to central regeneration failure. Along these lines, the authors argue that CNS axons such as corticospinal axons lose the ability to regenerate even in a permissive environment, such as a peripheral nerve graft. A recent report in Nat Med 2016 (Kadoya) showed that adult corticospinal axons do indeed regenerate into a permissive environment of a neural stem cell graft. This undermines the authors' argument. This by no means negates the authors' findings or the importance of their work, but perhaps they should present their findings in a modified context: overcoming maturation-associated alterations in axonal transport can enhance central axonal regeneration.

We agree with these comments, and we have changed wording throughout the paper to make it clear that we are describing one of several mechanisms- these changes are in red. However, we believe that many of the mechanisms controlling regeneration will be seen to depend on the mechanisms controlling transport and trafficking such we have described. This is a new focus for the field, and the data on rab11 that we give in this paper is just one of several transport and trafficking mechanisms that control axon growth and regeneration. We have also commented on axonal sprouting and regeneration into embryonic grafts. Our view is that there must be one or more unidentified axonal surface receptors that interact with ligands in the adult and embryonic CNS environment, that are responsible for these behaviours, and which are transported into mature CNS axons (presumably independently of rab11). It would be a great advance to identify these receptors.

The following is in two places in the discussion.

“However, some regenerative events are seen after CNS axotomy in adulthood. In the corticospinal tract and other pathways extensive lateral sprouting can occur (Bareyre et al., 2004), particularly in primates (Rosenzweig et al., 2010), and growth into embryonic grafts can occur (Bernstein-Goral and Bregman, 1993; Kadoya et al., 2016).”

The changes in transport that we have shown affect regeneration, but CNS axons nevertheless show some regenerative activity after damage, particularly lateral sprouting and growth into embryonic tissues. It is unlikely that these events can happen without the presence of a receptor on the axons and a corresponding ligand in the environment. These receptors, currently unidentified, must continue to be transported into mature axons, presumably independently of rab11.2) The authors might comment on in vivo extension of their work. Did the authors already attempt to extent these tools to in vivo models and encountered some limitations? In my view, this would not at all undermine the importance of this work or its suitability for publication. I also appreciate that in vivo experiments can fail for many reasons unrelated to the accuracy of the proposed mechanism, and that practical limitations in vivo may preclude clear documentation of an in vitro finding that is well documented, as in the case of the present results. The authors might at least discuss in vivo extension of their work, and issues that might be encountered.

We do not believe that overexpression of rab11 will promote long tract regeneration in vivo, and we do not plan to try it. The mistrafficking of rab11 and associated integrin into axons is to the proximal region and for in vivoregeneration we need a longer range effect. The data in this paper are designed to help us analyse the loss of regeneration with maturity, and provide proof of principle for a role of rab11 vesicles and their contents in regeneration. However we are working on solutions based on these observations that should promote CNS regeneration. We find that axonal transport of rab11 and integrins is controlled by Arf6 GAPs and GEFs, which can have a large effect on regeneration. A submitted paper is on http://biorxiv.org/content/biorxiv/early/2017/06/14/150037.full.pdf

We also have very strong regeneration-promoting effects and transport-promoting effects from overexpression of active PI3K isoforms, and from expression of an active form of protrudin. Our strategy is to find an optimum combination of interventions using the model described in this paper, then do a definitive in vivoexperiment with full behaviour. We do not like doing quickie regeneration experiments.

We have put the following comment in the discussion.

“Our results fit with this story, because PIP3 has effects on transport through GAPs and GEFs and on membrane trafficking (Macia et al., 2008; Randazzo et al., 2001). It is probable that there is a cycle, where transport of receptors allows PI3K activation and PIP3 generation, which in turn enhances transport and trafficking. Manipulation of rab11 transport using these methods can enhance regeneration in mature cortical neurons (Unpublished observations), and will be the basis of future in vivoinvestigations.”

3) The authors are using y-axes on a log scale in some of the later figures, likely to deal with a large spread in the data. This hinders however the ability to compare data presented in other panels on a linear scale. Please reconsider whether a log10 scale is the only way to present these data.

The distribution of this data is log Gaussian, so the clearest way to show clearly that the axon retraction after axotomy divides into short retraction in immature axons, long retraction in mature axons, and a mixture of the two in maturing axons is to plot on a log scale. The data is also shown on a linear scale in the supplementary figure attached to Figure 4. To guide readers we have added the following in the Results section:

The same data are plotted on a non-log scale in Figure 4—figure supplement 1)

4) Although the study showed that over-expression of rab11a in DIV 16 neurons significantly enhanced some of their regenerative responses to axotomy, such as decreased long retract and increased regeneration length in two hours, it is unclear how rab11 achieved such promoting effects. For instance, it is not clear why rab11a had different trafficking directions and velocities in axons and dendrites. It is also not clear why rab11a overexpression-induced mis-trafficking into axons enhanced some regenerative responses to axotomy. Can the authors show that over-expression of rab11a results in increased transport of axon growth-related molecules into the axon? Conversely, will the known axon regeneration promoting approaches affect the rab11 trafficking or localization in aged neurons? The answers to these questions will greatly strengthen the proposed hypothesis.

The penultimate sentence contains an important suggestion, which we had begun to address when Dr. Koseki had to return to Japan. We have now finished an experiment which provides this new data, and the results certainly strengthen the paper.

Rab11 vesicles transport many molecules, many of them cell surface receptors involved in axon growth. We decided to see whether rab11 forced into mature axons would carry with it an integrin. Alpha5 integrin is expressed in the cortical neurons, and normally completely excluded from the axons. Integrins are powerful promoters of axon growth. We transfected neurons with rab11-GFP, leading to rab11 being present in the proximal axons (identified by neurofascin staining). We stained the neurons for endogenously produced alpha5 integrin, which is normally completely excluded from axons. We found that, as predicted, the rab11 had carried alpha5 integrin with it into the axons. This experiment is now described in subsection “rab11 positive vesicles are excluded from axons in mature neurons” and Figure 9, and a section of the Materials and methods section.

As to why overexpression induces mis-trafficking, there are several possibilities. Mis-trafficking is a common finding after overexpression of molecules. Two probable explanations are a) that the selective anterograde transport selection process is overloaded and cannot cope with the excess cargo, and b) that the overexpressed molecule is excluded from axons by retrograde transport out of axon of molecules that have diffused into the axons. We think the second is more likely. We have put a sentence into the discussion:

As with many molecules the selective transport mechanism is somewhat leaky, which could indicate passive diffusion of vesicles and molecules into axons followed by selective removal by retrograde transport.

The final sentence asks whether known regeneration-promoting approaches affect transport. The paper answers this question for immature vs. mature axons. We are currently working on PIP3 levels, modulated by overexpression of PI3K. One isoform of PI3K has a very strong effect on both anterograde transport of rab11 and integrins and on regeneration. We plan to write this up soon.

The mechanisms by which selective transport is regulated in axons, and methods to alter it and promote regeneration are a main topic of current research in our lab. One mechanism is described in the paper on http://biorxiv.org/content/biorxiv/early/2017/06/14/150037.full.pdf. We are also working on regulation of transport and regeneration by PI3K/PIP3, protrudin and initial segment knockdown.